# GeoLink: Empowering Remote Sensing Foundation Model with OpenStreetMap Data

**Lubin Bai[1]***, **Xiuyuan Zhang[2]**, **Siqi Zhang[3]**, **Zepeng Zhang[4]**,
**Haoyu Wang[2]**, **Wei Qin[1]**, **Shihong Du[2]†**

[1] School of Earth and Space Sciences, Peking University, Beijing, China
[2] College of Urban and Environmental Sciences, Peking University, Beijing, China
[3] State Key Laboratory of Multimodal Artificial Intelligence Systems
Institute of Automation, CAS, Beijing, China
[4] Intelligent Maintenance and Operations Systems Lab
École Polytechnique Fédérale de Lausanne, Lausanne, Switzerland

## Abstract

Integrating ground-level geospatial data with rich geographic context, like OpenStreetMap (OSM), into remote sensing (RS) foundation models (FMs) is essential for advancing geospatial intelligence and supporting a broad spectrum of tasks. However, modality gap between RS and OSM data, including differences in data structure, content, and spatial granularity, makes effective synergy highly challenging, and most existing RS FMs focus on imagery alone. To this end, this study presents GeoLink, a multimodal framework that leverages OSM data to enhance RS FM during both the pretraining and downstream task stages. Specifically, GeoLink enhances RS self-supervised pretraining using multi-granularity learning signals derived from OSM data, guided by cross-modal spatial correlations for information interaction and collaboration. It also introduces image mask-reconstruction to enable sparse input for efficient pretraining. For downstream tasks, GeoLink generates both unimodal and multimodal fine-grained encodings to support a wide range of applications, from common RS interpretation tasks like land cover classification to more comprehensive geographic tasks like urban function zone mapping. Extensive experiments show that incorporating OSM data during pretraining enhances the performance of the RS image encoder, while fusing RS and OSM data in downstream tasks improves the FM's adaptability to complex geographic scenarios. These results underscore the potential of multimodal synergy in advancing high-level geospatial artificial intelligence. Moreover, we find that spatial correlation plays a crucial role in enabling effective multimodal geospatial data integration. Code, checkpoints, and using examples are released at https://github.com/bailubin/GeoLink_NeurIPS2025

## 1 Introduction

Remote sensing (RS) serves as a powerful tool for observing and monitoring our planet. Recently, the label-free, task-agnostic nature of self-supervised learning (SSL) has enabled RS foundation models (FMs) to make significant strides [1, 2, 3, 4]. Beyond scaling up the model parameters and dataset size, many RS FMs have been specifically tailored to accommodate the unique characteristics of RS image [5, 6, 7, 8], incorporating multi-scale [2, 9], multi-temporal [4, 10, 11, 12, 13], and multi-spectral [3, 14, 15, 16] processing techniques. After pretraining, these FMs can extract

---

*Email: lbbai@stu.pku.edu.cn
†Corresponding author: shdu@pku.edu.cn

39th Conference on Neural Information Processing Systems (NeurIPS 2025).

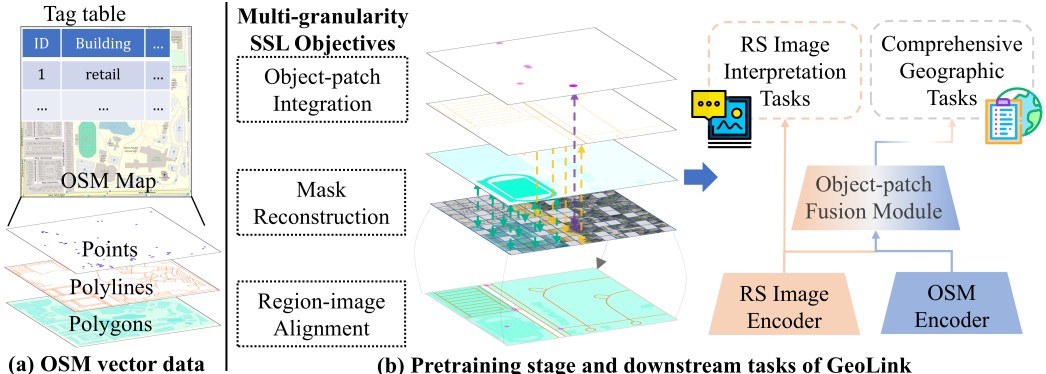

**(a) OSM vector data**      **(b) Pretraining stage and downstream tasks of GeoLink**

Figure 1: (a) OSM data stores the geometry information of geographic features in vector format, including points, polylines, and polygons, and leverages tags to record the semantic information. (b) GeoLink leverages multi-granularity SSL objectives to integrate RS and OSM data across multiple spatial scales, supporting both RS interpretation tasks and comprehensive geographic tasks.

meaningful, generalizable representations for RS interpretation tasks like semantic segmentation, yielding impressive performance in many domains, like environment monitoring [17] and disaster management [1, 18].

However, the integration of ground-level geospatial data remains relatively underexplored in many existing RS FMs. Ground-level geospatial data like various kinds of maps, in-situ sensor data and so on, can not only serve as RS interpretation references but also provide supplementary information for real-world applications [19, 20, 21]. Among them, OpenStreetMap (OSM) is one of the largest open-source geospatial databases of volunteered geographic information (VGI), providing rich geo-context associated with geographic locations [22]. OSM data has long been used in RS interpretation [20, 23, 24], and we believe integrating it into RS FM is essential for achieving a geo-oriented and context-aware understanding of Earth observation, as well as the advanced geospatial intelligence. First, it provides explicit location-based contextual cues that are difficult to capture from pure visual analysis, linking pixels to real-world objects and resolving ambiguities (e.g., distinguishing similar-looking buildings by location). Second, vision-language models like CLIP [25] show that structured semantics can enhance transferable representation learning, where OSM data, with its spatial hierarchies and geo-tagged attributes, plays a similar role. Finally, many geographic tasks demand a holistic understanding that RS images alone cannot provide, as they lack socioeconomic insights, while OSM data can fill this information gap.

As shown in Fig. 1(a), OSM data is originally vector-based, storing geographic objects as points, polylines, and polygons with rich tag tables, which differs significantly from RS image in data format and information content. To support RS image interpretation, most existing studies adopt indirect integration strategies. For example, converting OSM data into labels for RS images [26, 27], or constructing knowledge graphs from OSM data to provide prior knowledge for RS interpretation [28, 29]. However, such approaches tend to be manual-intensive and task-specific, confined to small-scale training datasets and experimental regions, making them misaligned with the paradigm of RS FMs. Some recent studies have explored leveraging OSM data to generate synthetic text data for RS vision-language FMs [30, 31], where relevant content descriptions associated with RS images are extracted from OSM data. They also follow an indirect way to reconcile the modality discrepancies between OSM and RS data, resulting in the loss of spatial information. To unlock OSM's potential for FM development, we aim to design a geo-spatially explicit approach that directly harnesses OSM's raw vector elements to inject geo-context into RS FMs, providing multi-perspective geographic priors while enhancing model capabilities across diverse geospatial tasks.

In this study, we introduce GeoLink, a multimodal FM that (1) enhances RS self-supervised pretraining through OSM-derived multi-granularity learning signals, (2) achieves efficient pretraining via masked input, and (3) increases the performance and diversity of downstream tasks via RS-OSM fusion. First, we design a heterogeneous graph neural network (GNN)-based OSM encoder that specifically addresses the geometric heterogeneity (points/polylines/polygons), non-Euclidean structure, and dynamic attribute tags of OSM data. Representing OSM objects as nodes and their spatial

relations as edges, the encoder performs message passing to generate both object-level (node) and region-level (graph) encodings for interaction with the RS image encoder. Second, as shown in Fig. 1(b), using location as the bridge to build multi-granularity spatial correlation, the cross-modal learning signals are derived at two levels: (1) region-image level alignment via contrastive learning with explicit spatial extent matching, and (2) object-patch level interaction through position-aware cross-attention, capturing implicit spatial associations while preserving spatial consistency for joint representation learning. Third, inspired by MAE [32] and FLIP [33], we randomly mask a large portion of image patches (e.g., 75%) during pretraining, feeding only the visible ones to accelerate training while maintaining accuracy improvements. Finally, after pretrained on 1.2 million sample pairs, we evaluate GeoLink in two task collections, as shown in Fig. 1(b), RS interpretation tasks (unimodal) and comprehensive geographic tasks (multimodal). According to the real-world application scenarios, several benchmarks are employed to assess our model across various domains, including land use/cover, agriculture, and urban planning. We find that incorporating OSM data during pretraining significantly enhances the RS image encoder's capacity, while fusing RS and OSM data in downstream tasks improves the FM's adaptability to complex geographic scenarios.

## 2  Related work

**RS FMs.** Thanks to advances in computational power and deep learning, RS FMs have rapidly evolved, trending toward multi-scale, multi-temporal, and multi-sensor designs tailored to the unique traits of RS imagery. Given the visual variance of geographic objects across resolutions, models like Scale-MAE [2] and Cross-Scale MAE [9] extend MAE with multi-scale augmentation and position embeddings to handle varying ground resolutions. As surface changes driven by seasons and human activity are common, multi-temporal RS is key for tasks like change detection and crop mapping. Approaches such as SeCo [11] use time-separated image pairs for contrastive learning, while SatMAE [4] introduces temporal embeddings to encode timestamps. Multi-sensor RS combines data from sources like multispectral and SAR to enrich downstream tasks, prompting FMs such as CROMA [3], DOFA [34], SeaMo [10], MMEarth [35], Skysense [36], and OmniSAT [37] to support multi-sensor inputs via multi-encoder or tokenizer-based designs. These works highlight the growing emphasis on multimodality. Beyond scale, time, and sensor diversity, geographic domain knowledge is also vital [38]. Geospatial vector data, such as OSM, offers rich yet underused geographic contextual information; this study explores its integration into RS FM.

**Synergy of RS and geospatial vector data.** A significant modality gap exists between RS and geospatial vector data, with most existing methods adopting indirect integration strategies. They can be categorized into three types based on vector data utilization: data conversion, data derivation, and knowledge graph methods. Data conversion methods utilize tools like buffering and rasterization to transform vector data into raster format, thus matching the structure of RS images for easier processing; the rasterized geospatial data may then serve as either inputs [39, 40] or training labels [22, 41, 42]. Data derivation methods generate intermediate data from vectors to assist RS tasks, such as producing image captions [30, 31], creating geospatial units from road networks [39], or constructing positive pairs for contrastive learning [43]. Knowledge graph-based methods extract geographic knowledge from vector data to build graphs that support RS image interpretation [28, 29]. While these indirect paradigms have long dominated RS-vector synergy, recent efforts explore direct integration, primarily via point data enriched with latitudinal-longitudinal priors [44, 45]. In this study, we aim to further harness the rich information contained in OSM vector data to incorporate with RS images, thereby improving performance in a wider array of geographic downstream tasks.

## 3  Method

**Framework Overview.** As shown in Fig. 2, GeoLink contains three encoders: (1) Vision Transformer (ViT)-based [46] RS image encoder $f_I$ that encodes RS image input $I$ into patch encodings $\varepsilon_P \in \mathbb{R}^{L_P \times D_P}$, where $L_P$ is the number of patches and $D_P$ is the patch feature dimension. (2) Graph Attention Convolution Network (GATConv)-based [47] OSM encoder $f_O$ that takes the constructed OSM graph $G$ as input, and outputs the node encodings after message passing $\varepsilon_V \in \mathbb{R}^{L_V \times D_V}$, where $\varepsilon_V = \varepsilon_{V_p} \cup \varepsilon_{V_l} \cup \varepsilon_{V_g}$. Here, $L_V$ is the number of nodes, $D_V$ is the node feature dimension, and $p$, $l$, and $g$ denote the node types: point, polyline, and polygon, respectively. Details about the OSM encoder structure can be found in the Appendix A. (3) Object-patch fusion encoder $f_F$

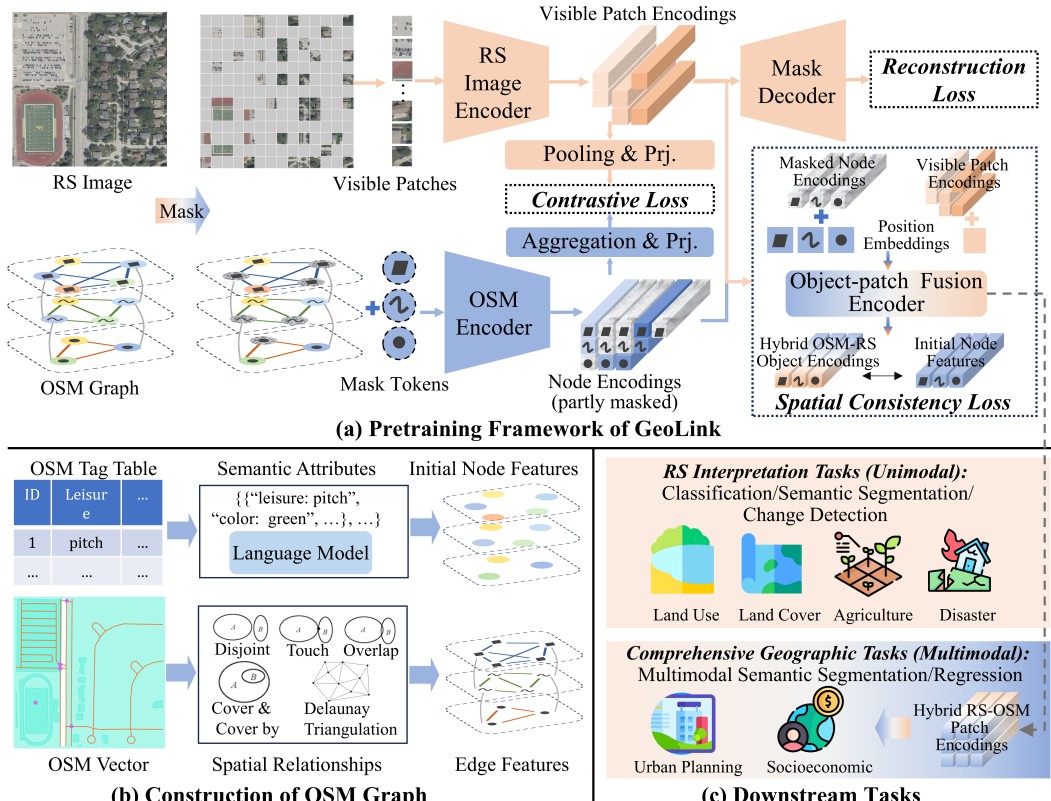

**(a) Pretraining Framework of GeoLink**

**(b) Construction of OSM Graph**

**(c) Downstream Tasks**

Figure 2: (a) GeoLink masks both modalities, using visible image patches and masked OSM graph as inputs. Pretraining is achieved through three SSL objectives: RS reconstruction loss, cross-modal contrastive loss, and spatial consistency loss. (b) The heterogeneous graph is employed to model OSM data, incorporating three node types and multiple spatial relationships. (c) The pretrained model can produce both unimodal and multimodal encodings, generalizing to various downstream tasks.

for fine-grained data integration, which takes the patch encodings $\varepsilon_P$ and node encodings $\varepsilon_V$ as input, and generates two types of multimodal encodings, including hybrid OSM-RS object encodings $\varepsilon_{OR} \in \mathbb{R}^{L_V \times D_F}$ and hybrid RS-OSM patch encodings $\varepsilon_{RO} \in \mathbb{R}^{L_P \times D_F}$, where $D_F$ is the fusion feature dimension. During pretraining, we mask both modalities, using visible RS image patches and masked OSM graphs as inputs. Three learning objectives are leveraged to optimize the encoders, including RS reconstruction loss, cross-modal contrastive loss, and spatial consistency loss. Next, we will systematically introduce GeoLink's pretraining process.

**OSM Graph Construction.** As shown in Fig. 2(b), the heterogeneous graph is constructed to model the OSM vector map, where nodes represent geographic objects (points, polylines, and polygons), and edges capture various spatial relationships between them. First, we embed the OSM tags as the initial node features to represent the semantic attributes. OSM follows a free tagging system with unlimited tag categories, rendering static methods inadequate for handling unseen values. To accommodate this scenario while preserving semantics, we employ a language model to encode the OSM tags. Specifically, each OSM object's tag attributes are organized into a tag-value dictionary $D_o = \{(t_1, c_1), (t_2, c_2), \ldots, (t_n, c_n)\}$, and each tag-value pair is converted into a string $s_i = \text{concat}(t_i, \texttt{":"}, c_i)$, which is individually encoded using a BERT language model as $h_i = \text{BERT}(s_i)$. More frequently occurring tags (e.g., "building", "land use") better represent the general attributes of an object, while rare tags (e.g., "historic period") usually describe details. To provide a comprehensive and consistent representation, we calculate the weighted average of all tag-value encodings of an OSM object to serve as the initial node feature:$\sigma_V = \sum\limits_{i=1}^{n} w_i h_i / \sum\limits_{i=1}^{n} w_i$, where the weight $w_i$ for each tag corresponds to its occurrence count in the global pretraining dataset. Second, given the scale differences between points, polylines, and polygons, we leverage spatial topological relationships rather than distances to construct edges, accounting for variations across

different vector types. For instance, point features only exhibit disjoint relationships, in which case we employ Delaunay Triangulation [48] to establish connections. A detailed exposition of the spatial relationship types among various node categories is provided in the Appendix A. Compared to spatial distance, topology embodies a definitive spatial relationship and is less susceptible to noise. The constructed OSM graph is then fed into the OSM encoder, where message passing enables each node to aggregate information from its neighbors, resulting the node encodings $\varepsilon_V$.

**Masked Inputs.** During the pretraining phase, the learning objectives are built upon masked inputs, which not only enhances representation learning but also reduces memory consumption. For RS data with a ViT encoder, the image is divided into a grid of non-overlapping patches, with a large portion randomly masked out, leaving only the visible ones as input to obtain the visible patch encodings $\varepsilon_P^v$. Following MAE [32], we leverage a mask decoder containing two Transformer blocks to reconstruct the masked patches. Then, we calculate the reconstruction loss between the reconstructed patches $\hat{I}^m$ and original masked patches $I^m$ as:

$$\mathcal{L}_{\text{rec}} = \frac{1}{N} \sum_{i=1}^{N} \frac{1}{L_P^m} \sum_{j=1}^{L_P^m} \left( \hat{I}_{ij}^m - I_{ij}^m \right)^2 ,$$

where $N$ is the batch size and $L_P^m$ is the number of masked patches in each input image. For OSM data, we employ a node-masking strategy. For each masked node $i$, its initial feature $\sigma_{V_i}$ is replaced by a learnable mask token while preserving its original adjacency edges with other nodes to ensure effective message passing within the OSM encoder. We adopt three different mask tokens for points, polylines, and polygons, respectively. The OSM encoder outputs encodings $\varepsilon_V$ for all nodes, including both masked and visible ones, i.e., $\varepsilon_V = \varepsilon_V^m \cup \varepsilon_V^v$.

**Region-image Level Alignment.** Since each input pair of RS and OSM data covers the same geographic extent, they describe the corresponding region from different perspectives and contain correlated, complementary information. Consequently, we employ contrastive learning to align them. To obtain region-level OSM encoding, we design an aggregation module to aggregate the heterogeneous node encodings. Specifically, this module first leverages three Set2Set layers that independently aggregate nodes of each type into type-specific encodings $\varepsilon_{G_t} = \text{Set2Set}_t(\varepsilon_{V_t})$, where $t \in \{p, l, g\}$ corresponds to the node types of point, polyline, and polygon. Then a linear layer followed by a Softmax layer computes type attention to weight-sum the type-specific encodings to produce the OSM region encoding: $\varepsilon_G = \sum_{t \in \{p,l,g\}} \text{Softmax}(\text{Linear}(\varepsilon_{G_t}))\varepsilon_{G_t}$. For RS data, we employ mean pooling for the visible patch encodings $\varepsilon_P^v$ to obtain the image-level encoding $\varepsilon_I = \text{MeanPool}(\varepsilon_P^v)$. Following [25], we project both the OSM region encoding $\varepsilon_G$ and the RS image encoding $\varepsilon_I$ using separate linear layers, i.e., $z_G = \text{Linear}_G(\varepsilon_G)$ and $z_I = \text{Linear}_I(\varepsilon_I)$, and contrast both modalities using the InfoNCE loss [49]:

$$\mathcal{L}_{\text{cont}} = -\frac{1}{2N} \left( \sum_{i=1}^{N} \frac{\exp(\text{sim}(z_G^i, z_I^i)/\tau)}{\sum_{j=1}^{N} \exp(\text{sim}(z_G^i, z_I^j)/\tau)} + \sum_{i=1}^{N} \frac{\exp(\text{sim}(z_I^i, z_G^i)/\tau)}{\sum_{j=1}^{N} \exp(\text{sim}(z_I^i, z_G^j)/\tau)} \right) ,$$

where $N$ is the batch size, $\tau$ is a temperature parameter, and $\text{sim}(\cdot, \cdot)$ is the cosine similarity function, i.e., $\text{sim}(u, v) = \frac{u^\top v}{\|u\|\|v\|}$. Through contrastive learning, the inherent structured semantic information in OSM data is effectively conveyed to the image encoder, thereby guiding its pretraining process.

**Object-patch Level Integration.** To further facilitate interaction between the two modalities and obtain fine-grained fused representations, we design an object-patch fusion encoder. This encoder is implemented as a two-way Transformer composed of one self-attention layer and two cross-attention layers (details in Appendix A). As shown in Fig. 2(a), during pretraining, the fusion encoder accepts the masked node encodings $\varepsilon_V^m$ and the visible patch encodings $\varepsilon_P^v$ as inputs, and produces two types of fused encodings: hybrid OSM-RS object encodings $\varepsilon_{OR}^m$ and hybrid RS-OSM patch encodings $\varepsilon_{RO}^m$. A critical challenge arises from the inherent spatial ambiguity between the two input encodings: the absence of explicit geographic correspondence makes direct cross-attention operations susceptible to erroneous feature associations, e.g., accidentally connect unrelated elements. To address this, we incorporate sinusoidal position embeddings into the fusion encoder by adding them to each input. The sinusoidal embedding handles individual coordinates, which can be directly adopted to point nodes. To capture the spatial coverage of polylines and polygons, we sample their key-points and compute the average of the sampled points' position embeddings (details see in Appendix A). These enhanced spatial signatures enable the model to progressively establish accurate cross-modal association

through attention-based learning. As shown in Fig. 2(a), the output hybrid OSM-RS object encodings integrate features from visible nodes (via message passing in the OSM encoder) and visible patches (via cross-modal interaction in the fusion encoder), which encapsulate the spatial context surrounding masked OSM objects. According to the first law of geography [50], this contextual information is strongly correlated with the intrinsic properties of the masked ones. To enforce spatial-semantic consistency, we introduce a consistency loss function that operates on the hybrid OSM-RS object encodings $\varepsilon_{OR}^m$ and the initial features of masked nodes $\sigma_V^m$:

$$\mathcal{L}_{\text{cst}} = \frac{1}{N} \sum_{i=1}^{N} \frac{1}{L_V^m} \sum_{j=1}^{L_V^m} \left( \varepsilon_{OR_{ij}}^m - \sigma_{V_{ij}}^m \right)^2,$$

where $N$ is the batch size, and $L_V^m$ is the number of masked nodes in each graph. The synergy of the position embedding and consistency constraints significantly enhances the model's capacity for cross-modal representation learning, improving the RS encoder's ability to capture fine-grained semantic features. Finally, the three objectives are combined together for pretraining: $\mathcal{L} = \alpha \mathcal{L}_{\text{rec}} + \beta \mathcal{L}_{\text{cont}} + \gamma \mathcal{L}_{\text{cst}}$, where the loss weights are set as $\alpha = 1$, $\beta = 0.01$, and $\gamma = 0.01$ by default (related experiments about the loss weights are provided in Appendix D).

## 4 Experiments

**Pretraining details.** To pretrain GeoLink, we construct a multimodal dataset derived from SkyScript-top30 [31]. The SkyScript-top30 dataset contains multi-source, multi-resolution RS images with RGB bands, featuring ground sample distances (GSD) ranging from 0.1 m/pixel to 30 m/pixel. For each RS image, the corresponding OSM data is downloaded from the Overpass API using its geo-coordinate and timestamp. After preprocessing like data cleaning, we obtain a final pretraining dataset of 1,271,431 matched pairs. More details are provided in the Appendix B. A ViT-L model is employed as the RS image encoder in this study. The default masking ratios for RS patch and OSM graph node is $75\%$ and $20\%$. And $\tau = 0.2$ for contrastive loss. Our experiments are conducted on a Linux server equipped with 4 NVIDIA RTX6000 GPUs (48GB) using bfloat16 precision. Unlike FMs such as Scale-MAE (800 epochs) and CROMA (600 epochs) which typically require a large number of pretraining epochs, GeoLink demonstrates significantly faster convergence. We pretrain it for only 60 epochs (including 5 warmup epochs), with a batch size of 2640, a base learning rate of $1 \times 10^{-4}$, and a cosine decay schedule for learning rate cooldown. For data augmentation, we apply random cropping, horizontal flipping, and color jittering to the RS images. Notably, we perform corresponding geometric transformations on the OSM data to maintain spatial alignment with the augmented RS images. The model optimization employs AdamW with hyperparameters ($\beta_1 = 0.9$, $\beta_2 = 0.95$) and a weight decay of 0.05.

**Downstream Task Settings.** As depicted in Figure 2(c), GeoLink supports both RS image interpretation tasks (unimodal) and comprehensive geographic tasks (multimodal). For the former, only the RS image encoder is employed, which is combined with task-specific protocols to assess the learned unimodal representations through classification, semantic segmentation, and change detection tasks. We benchmark GeoLink against six RS FMs, including GASSL [45], MMEarth [35], Scale-MAE [2], Cross-scale MAE [9], CROMA [3], and DOFA [34]. Other comparisons can be found in the Appendix D. For multimodal geographic tasks, we incorporate OSM data into the downstream tasks. We evaluate on urban function zone (UFZ) segmentation, urban village (UV) identification, population density (POP) and carbon emission (CO2) estimation tasks, which are crucial for realistic urban planning and socioeconomic analysis. These challenges typically require multi-source geographic data, making them ideal for assessing GeoLink's multimodal capabilities. The details of baseline model selection, downstream model architectures, and evaluation protocols are provided in the Appendix C.

### 4.1 Unimodal tasks

**Classification tasks.** We evaluate the RS representations learned by GeoLink using three classification protocols, including: (1) kNN ($k = 20$) evaluates representation quality by measuring instance clustering without further training; (2) linear probing assesses performance using a linear classifier on frozen RS features; (3) fine-tuning jointly updates the RS encoder and classifier for task-specific adaptation. For comprehensive evaluation, we employ seven RS benchmarks that span diverse spatial

Table 1: Comparison of different models across seven classification benchmarks under kNN, linear probing (LP), and fine-tuning (FT) evaluation protocols (Top-1 accuracy %).

| Model | Backbone | MLRSNet | | | EuroSAT | | | WHU-RS19 | | | OPTIMAL-31 | | | RESISC-45 | | | AiRound | | | UCMerced | | |
|---|---|---|---|---|---|---|---|---|---|---|---|---|---|---|---|---|---|---|---|---|---|---|---|
| | | kNN | LP | FT | kNN | LP | FT | kNN | LP | FT | kNN | LP | FT | kNN | LP | FT | kNN | LP | FT | kNN | LP | FT |
| GASSL | ResNet50 | 91.28 | 93.08 | 95.83 | 91.24 | 94.13 | 97.34 | 86.88 | 96.62 | 96.82 | 76.56 | 85.48 | 86.77 | 81.50 | 87.19 | 92.78 | 65.52 | 75.47 | 77.66 | 82.67 | 93.62 | 95.14 |
| MMEarth | ConvNext V2 | 89.29 | 91.42 | 96.31 | 94.84 | 96.42 | 98.27 | 90.82 | 96.18 | 97.72 | 73.11 | 89.74 | 90.64 | 80.21 | 88.50 | 94.44 | 68.30 | 73.65 | 75.68 | 86.67 | 95.36 | 96.03 |
| Scale-MAE | ViT-L | 92.26 | 93.56 | 96.97 | 93.42 | 97.40 | 98.06 | 90.85 | 98.41 | 98.01 | 78.28 | 87.63 | 88.71 | 85.42 | 91.14 | 94.15 | 71.14 | 78.20 | 82.84 | 78.38 | 96.10 | 95.71 |
| Cross-Scale MAE | ViT-L | 92.63 | 93.30 | 96.23 | 93.24 | 95.58 | 97.97 | 89.66 | 97.02 | 97.42 | 79.78 | 87.85 | 88.71 | 85.10 | 90.89 | 93.54 | 72.23 | 74.81 | 80.79 | 84.67 | 95.90 | 96.19 |
| CROMA | ViT-L | 89.95 | 92.64 | 96.21 | 94.64 | 97.13 | 98.17 | 85.88 | 95.43 | 96.22 | 77.85 | 85.05 | 86.88 | 82.41 | 88.61 | 93.36 | 65.40 | 73.33 | 80.52 | 81.90 | 93.81 | 94.58 |
| DOFA | ViT-L | 90.73 | 92.40 | 96.36 | 93.93 | 96.79 | 98.20 | 90.04 | 98.12 | 98.31 | 77.28 | 90.89 | 90.54 | 83.04 | 89.85 | 93.85 | 70.27 | 75.48 | 78.35 | 87.41 | 95.16 | 96.50 |
| GeoLink | ViT-L | 93.48 | 93.49 | 97.35 | 95.22 | 97.30 | 98.30 | 91.05 | 98.81 | 98.41 | 82.37 | 91.40 | 91.72 | 87.33 | 91.42 | 94.45 | 72.52 | 77.59 | 83.38 | 87.43 | 98.19 | 98.10 |

Table 2: Comparison of encoder-freezing and fine-tuning performance across four semantic segmentation/change detection downstream datasets (mIoU %).

| Model | Backbone | FiveBillionPixels | | AI4Smallfarms | | SpaceNet7 | | xView2 | |
|---|---|---|---|---|---|---|---|---|---|
| | | Freezing | Fine-tuning | Freezing | Fine-tuning | Freezing | Fine-tuning | Freezing | Fine-tuning |
| GASSL | ResNet50 | 57.47 | 61.37 | 39.65 | 43.29 | 57.63 | 62.09 | 56.27 | 59.87 |
| MMEarth | ConvNext V2 | 56.12 | 62.69 | 37.86 | 43.13 | 62.20 | 62.66 | 56.54 | 59.92 |
| Scale-MAE | ViT-L | 58.94 | **65.73** | 41.11 | 45.98 | 62.78 | 63.22 | 58.42 | 60.37 |
| Cross-Scale MAE | ViT-L | 58.68 | 63.96 | 40.33 | 44.87 | 60.23 | 63.03 | 58.44 | 60.87 |
| CROMA | ViT-L | 58.09 | 63.96 | 41.16 | 45.89 | 58.97 | 61.19 | 57.34 | 59.12 |
| DOFA | ViT-L | 57.83 | 63.94 | 38.31 | 45.94 | 61.38 | 62.43 | 58.86 | 61.47 |
| GeoLink | ViT-L | **60.49** | 64.93 | **43.26** | **47.29** | **63.22** | **64.07** | **59.94** | **61.94** |

resolutions and category systems: MLRSNet [51], EuroSAT [52], WHU-RS19 [53], OPTIMAL-31 [54], RESISC-45 [55], AiRound [56], and UCMerced [57]. All benchmarks are split into 50% for training, 10% for validation, and 40% for testing. Each experiment is repeated three times under different random seeds and the average results are reported to ensure robustness. Please refer to the Appendix C for detailed downstream task settings. As shown in Table 1, GeoLink achieves state-of-the-art performance on most datasets, showcasing its superiority in learning generalizable representations. Notably, GeoLink outperforms all compared FMs by significant margins under the kNN protocol, which indicates that it has learned structured RS representations where semantically similar samples are close in the feature space. Compared to linear probing, GeoLink demonstrates even more pronounced superiority under the fine-tuning protocol. Beyond this, we also conduct data efficiency analysis, and observe that GeoLink's advantage becomes even more evident when training samples are limited. Detailed results are provided in Appendix D.

**Semantic segmentation and change detection tasks.** Unlike classification tasks, semantic segmentation and change detection aim to evaluate the model's ability to capture spatially detailed representations. For both tasks, we follow the protocols of the PANGAEA-bench, including using UperNet as the decoder and adopting identical training configurations. We evaluate performance on four benchmarks from PANGAEA-bench: Five-Billion-Pixels [58], AI4SmallFarms [59], xView2 [60], and SpaceNet7 [61], which cover diverse application domains such as agricultural monitoring and disaster management. The first two benchmarks correspond to semantic segmentation tasks, while the latter two are used for change detection. Table 2 presents the performance comparison of various FMs on the four benchmarks under both encoder-freezing and fine-tuning settings. GeoLink consistently achieves the best results on average in both scenarios, demonstrating the strong generalization and adaptability of its learned RS representations. The advantages of GeoLink are also evident in more challenging datasets such as AI4Smallfarms and xView2, where it consistently leads under both settings. When considered alongside the unimodal classification results, these findings further underscore the effectiveness of GeoLink's cross-modal pretraining strategy. By leveraging the supervision from OSM data, GeoLink significantly enhances the capability of the RS image encoder to learn transferable and semantically rich representations.

### 4.2 Multimodal tasks

**UFZ segmentation.** UFZs, such as residential, commercial, and institutional areas, are essential units in urban planning, reflecting complex socioeconomic and physical dynamics. However, due to the heterogeneous nature of man-made infrastructure and visual discontinuities in RS images, accurately identifying UFZs using RS data alone remains challenging. To address this, growing research efforts incorporate multi-source data [62, 63, 64, 65]. We construct a challenging real-world UFZ segmentation benchmark by refining planning maps from Chicago, Singapore, and Shenzhen, comprising 60,970 samples across nine UFZ categories. Using UperNet as the decoder, we fine-tune

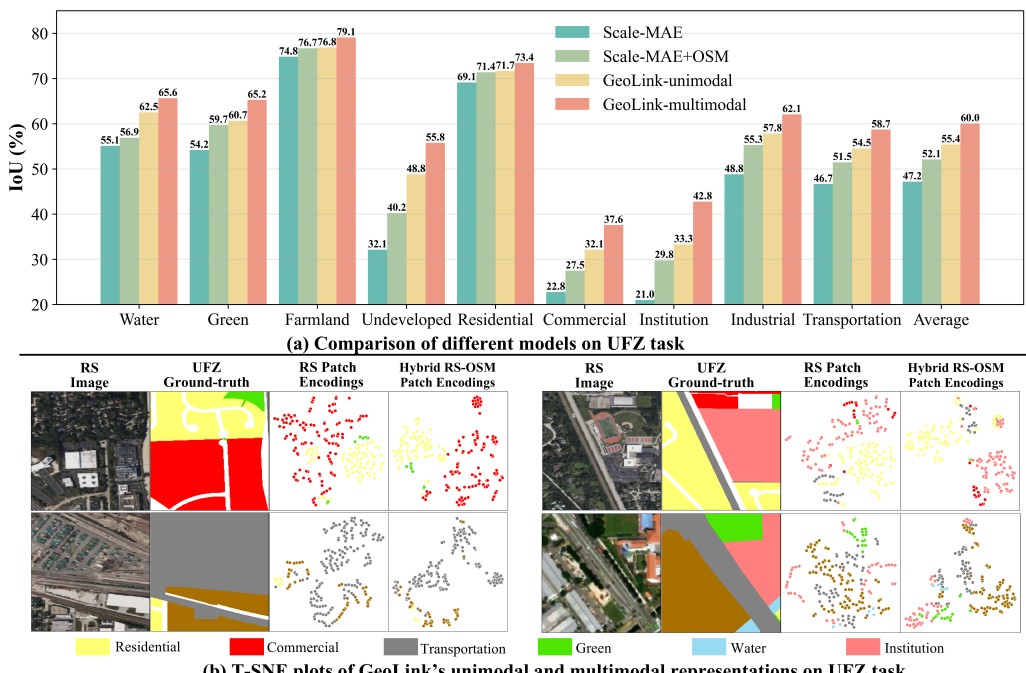

**(a) Comparison of different models on UFZ task**

**(b) T-SNE plots of GeoLink's unimodal and multimodal representations on UFZ task**

Figure 3: (a) IoU (%) performances of each UFZ category. (b) T-SNE is used to visualize the learned patch encodings of GeoLink. With the incorporation of OSM data, multimodal encodings become more compact and discriminative than unimodal ones.

GeoLink and Scale-MAE under unimodal and multimodal settings. In the multimodal setup, UperNet consumes hybrid RS-OSM features produced by the object-patch fusion encoder. Detailed structures and settings are provided in the Appendix C. GeoLink outperforms Scale-MAE in both unimodal and multimodal scenarios (Fig. 3(a)). Notably, when OSM data is excluded at downstream task time, GeoLink benefits from multimodal pretraining, yielding superior performance. When OSM data is included, GeoLink's performance further improves—particularly for complex classes such as industrial, institutional, and commercial—highlighting the advantages of precise, semantically rich multimodal fusion. In addition, Scale-MAE+OSM largely outperforms Scale-MAE, further indicating the effectiveness of the proposed designed fusion encoder. Furthermore, we visualize the patch representations of GeoLink in Figure 3(b). Some categories that are easily confused in RS images exhibit scattered and overlapping distribution in unimodal scenario, while the hybrid RS-OSM patch encodings show significantly greater discriminability, highlighting GeoLink's multimodal fusion capability and the importance of multi-source information in comprehensive geospatial tasks. We also conduct a relevant while distinctive downstream task, i.e., urban village identification, and detailed results are provided in Appendix D.

**UV identification.** UVs—informal residential zones within or on the outskirts of cities—often suffer from deficient infrastructure and low-quality living conditions. Identifying UV is important for urban planning and sustainable development. In this study, we construct an UV semantic segmentation dataset (details in Appendix C) to evaluate the learned representations.

Table 3: Comparison of methods on the urban village identification task (IoU %).

| Method | Background | UV | mIoU |
|---|---|---|---|
| Scale-MAE | 90.29 | 58.21 | 74.25 |
| Scale-MAE+OSM | 91.37 | 68.81 | 80.09 |
| GeoLink-unimodal | 90.40 | 68.67 | 79.53 |
| GeoLink-multimodal | **92.29** | **71.08** | **81.68** |

The results in Table 3 demonstrate that GeoLink maintains outstanding performance in this task, showing significant improvement compared to Scale-MAE whether in unimodal or multimodal scenarios. Comprehensively analyzing both the UV and UFZ tasks, the hybrid RS-OSM patch representations generated by GeoLink effectively couple information from both data sources, making them suitable for fine-grained tasks like semantic segmentation.

**POP and CO2 estimation.** Fine-scale spatialized population and carbon emission data are essential for geoscience research, including climate change studies. In this work, we evaluate GeoLink on both tasks. Grid-based population density and carbon emission data for Chicago, Singapore, and Shenzhen are sourced from WorldPop and ODIAC to construct evaluation benchmarks (details in Appendix C). We use a two-layer MLP as the regression head for each task. As illustrated in Table 4, integrating OSM data significantly improves GeoLink's performance on both POP and CO2 estimation, highlighting the value of multi-source geospatial

Table 4: Performance comparison on POP and CO2 estimation tasks ($r^2$ %).

| Model | POP | CO2 |
|---|---|---|
| Scale-MAE | 47.18 | 59.12 |
| Scale-MAE+OSM | 48.29 | 59.97 |
| GeoLink-Unimodal | 49.76 | 62.37 |
| GeoLink-Multimodal | **51.88** | **65.12** |

data. This supports our view that multimodal data fusion will be essential to the next generation of FMs in geography.

## 4.3 Ablation studies

**Model designing.** Drawing on GeoLink's characteristics, we conduct comprehensive ablation studies to isolate the effects of key design choices, including learning objective, masking ratio, and position embedding for data fusion. Unless otherwise noted, every variant is pretrained for 60 epochs using ViT-L as the image encoder. We assess each configuration on kNN classification, linear probing, and UFZ segmentation. For kNN and linear probing, results are reported as the mean performance over seven benchmarks. The results are shown in Table 5.

(1) Learning objective. When using only $\mathcal{L}$rec, the model degenerates into a standard MAE. Under this setting, performance is notably poor after 60 epochs of pretraining and only becomes competitive after 300 epochs, indicating slow convergence. In contrast, GeoLink achieves strong performance within just 60 epochs, thereby reducing computational cost. Additionally, $\mathcal{L}$cont facilitates cross-modal alignment to enhance the image encoder, while $\mathcal{L}_{\text{cst}}$ primarily serves to optimize the fusion module to obtain multimodal encodings.

(2) RS masking ratio. Reducing the masking ratio to 50% for RS images leads to performance degradation and increased training cost, and we can also observe a slight performance drop at 80% and a significant collapse at 90%.

(3) OSM masking ratio. The model shows robustness to the masking ratio applied to OSM inputs, maintaining stable performance when approximately 20% of the data is masked.

(4) Position embedding for data fusion. Removing position embeddings causes a slight drop in unimodal performance but leads to a significant decline in multimodal effectiveness. This underscores the critical role of spatial correlation in the fusion of geospatial modalities. More experiments on the designing of position embedding can be found in the Appendix D.

Table 5: Ablation study on GeoLink. Performance on classification tasks (top-1 accuracy %) and UFZ segmentation (mIoU %) under varying configurations.

| Ablation | Setting | Cost | kNN | Linear | UFZ-U | UFZ-M |
|---|---|---|---|---|---|---|
| Default | – | 1.00× | 87.06 | 92.60 | 55.40 | 60.00 |
| (1) Learning objective | $\mathcal{L}_{\text{rec}}$ (60 epochs) | 0.85× | 78.12 | 83.42 | 40.41 | – |
| | $\mathcal{L}_{\text{rec}}$ (300 epochs) | 4.25× | 85.97 | 90.03 | 49.09 | – |
| | $\mathcal{L}_{\text{rec}} + \mathcal{L}_{\text{cont}}$ | 0.92× | 86.12 | 91.14 | 53.94 | – |
| | $\mathcal{L}_{\text{rec}} + \mathcal{L}_{\text{cst}}$ | 0.93× | 82.34 | 86.77 | 51.06 | 57.39 |
| (2) RS masking ratio | 50% | 2.10× | 83.21 | 89.64 | 49.96 | 54.28 |
| | 80% | 0.95× | 85.21 | 90.17 | 52.01 | 56.04 |
| | 90% | 0.90× | 78.02 | 83.47 | 43.02 | 45.81 |
| (3) OSM masking ratio | 15% | 1.00× | 86.99 | 91.97 | 54.21 | 58.70 |
| | 25% | 1.00× | 86.43 | 91.04 | 53.84 | 58.02 |
| (4) Fusion position embedding | Without | 1.00× | 86.47 | 91.87 | 53.97 | 56.22 |
| (5) OSM encoder variants | GCN variant | 1.00× | 87.01 | 92.14 | 55.02 | 59.42 |
| | Transformer variant | 1.01× | 86.14 | 91.47 | 53.73 | 58.57 |

(5) OSM encoder varians. We design two variant experiments to illustrate the effectiveness of the current OSM encoder. One is GCN variant which replaces the GAT in the OSM encoder with GCN while keeping all other settings unchanged, and the other is transformer variant which replaces the existing GNN-based OSM encoder with 2 standard Transformer blocks. To adapt to the Transformer's structure, each OSM node in the original GNN is treated as a token input to the Transformer. Additionally, position embedding is added to each token to preserve spatial information. Compared to the GCN variant, the attention mechanism introduced by GAT yields better performance. The Transformer variant underperforms both GCN and GAT, which we speculate is primarily due to the insufficient message-passing between OSM nodes in it, hindering the learning of OSM spatial consistency objective during pretraining.

**OSM data completeness.** Given the crowd-sourced nature of OSM, data completeness varies across regions, requiring models to remain robust under sparse conditions. To evaluate this, we simulate completeness by randomly removing OSM objects (Table 6). Results indicate only a minor performance drop when 20% of OSM data is removed. Even with 50% of the OSM data omitted, GeoLink maintains strong performance across most tasks, with a notable decline observed primarily on multimodal UFZ segmentation task. These findings suggest that GeoLink demonstrates substantial robustness to incomplete OSM coverage and can adapt to regions with limited OSM availability.

Table 6: Results of kNN (top-1 accuracy %) and UFZ (mIoU %) under varying OSM completeness.

| Completeness | kNN | UFZ-U | UFZ-M |
|---|---|---|---|
| 100% | 87.06 | 55.40 | 60.00 |
| 80% | 86.98 | 55.01 | 59.13 |
| 50% | 86.59 | 54.12 | 57.37 |

## 5   Conclusion

In this study, we propose GeoLink, which effectively integrates geographic contextual cues from OSM data into the RS FM through semantic-spatial feature extraction and spatial-aware cross-modal interaction. This design enhances the pretraining process of RS image encoder, significantly improving its performance on image interpretation tasks. In addition, GeoLink produces fine-grained hybrid RS-OSM patch encodings tailored for comprehensive geographic tasks. Extensive evaluations across diverse benchmarks demonstrate that GeoLink outperforms previous state-of-the-art models and excels in more challenging downstream tasks such as UFZ mapping. Furthermore, our findings emphasize the pivotal role of spatial correlation in bridging and fusing multimodal geospatial data.

Despite its promising results, GeoLink has certain limitations: (1) The focus of this paper is to explore how to leverage the rich geographic information of OSM data in RS FM, and at present, it only supports RGB images and cannot process multispectral RS data. We believe the inclusion of multispectral RS images could further enhance the model, and we plan to improve the image encoder to accommodate data from various sensor types. (2) The current position embedding for OSM vectors can lead to the loss of spatial details. We argue that if position embedding can better capture the spatial characteristics, it can facilitate more accurate and deeper spatial correlations, thereby enhancing the synergy across multimodal geospatial data.

## 6   Acknowledgement

The work is funded by the National Key Research and Development Program of China (2023YFC3804802) and National Natural Science Foundation of China (No. 42330103). Shihong Du is the corresponding author.

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

# A    Details of model designing

## A.1    Spatial relationships contained in each edge type

Due to the different geometric structures of points, polylines, and polygons, the potential spatial relationships among these three node types also vary. Accordingly, the OSM heterogeneous graph we constructed incorporates nine edge types based on the permutations of different node types. As shown in Table 7, we summarize the spatial relationships contained by each edge type in detail. Specifically, while relationships between point nodes are represented using Delaunay triangulation, those among other node types are defined in terms of topological relations. After computing these spatial relationships, we encode them using one-hot encoding to represent the edge attributes within the heterogeneous graph.

Table 7: Topological relations between different node types

| Node type | Point | Polyline | Polygon |
|---|---|---|---|
| Point | Delaunay triangulation | Touch, within | Touch, within |
| Polyline | Touch, contain | Touch, intersect, cover, cover by, equal | Touch, cross, cover by |
| Polygon | Touch, contain | Touch, cross, cover | Touch, overlap, cover, cover by, equal |

## A.2    The structure of OSM encoder

The OSM encoder is a lightweight heterogeneous GNN built upon GATConv, specifically designed to capture the spatial and semantic relationships embedded in OSM data. As illustrated in Fig. 4, the encoder comprises nine parallel GATConv layers, each dedicated to handling one of nine distinct edge types defined by the spatial relationships between node types. The encoder supports three node types—point, polyline, and polygon—allowing it to model the complex topology inherent in OpenStreetMap (OSM) data. Taking point-type nodes as an example, the encoder not only performs intra-type message passing (i.e., among points) but also enables inter-type interactions by propagating messages to polyline and polygon nodes through separate GATConv layers.

Conversely, it also gathers contextual information from polyline and polygon neighbors using additional GATConv layers. This cross-type message exchange ensures that each node embedding is enriched by both its own type's local structure and the information from other geometric types. By explicitly modeling heterogeneous relations such as touches, within, crosses, and contains, the encoder effectively captures both the geometric connectivity and high-level geographic semantics of OSM features. The resulting node embeddings provide a rich representation of the spatial environment, which can be further integrated with RS encodings for downstream tasks.

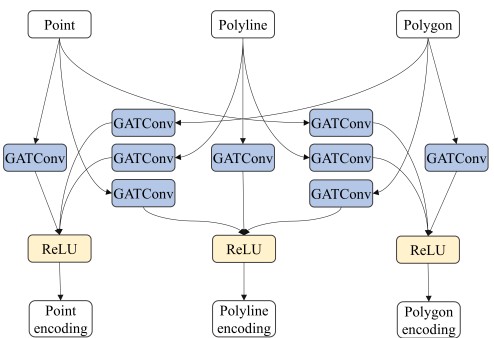

Figure 4: Detailed structure of the OSM encoder.

## A.3    Position embedding of OSM objects

We use sinusoidal position embedding to uniformly represent the spatial positions of points, polylines, polygons, and image patches, thereby implicitly establishing spatial associations for fine-grained RS-OSM fusion. As mentioned in Sec. 3, we sample key-points and compute the average of the sampled points' position embeddings to capture the spatial coverage of polylines and polygons during the object-patch level integration. For a polyline vector, we sample three key-points, i.e., two endpoints and midpoint. For a polygon vector, we first compute its centroid. Then, using the centroid as the center, we sample three points inside the polygon at random radii and angles, resulting in four points to represent its position and coverage. The sinusoidal position embedding is computed for each

sampled key-point, and their average is taken as the final position embedding for the corresponding vector. In Appendix D, we evaluate the effect of the sampling number of key-points on model performance.

## A.4 The structure of object-patch fusion encoder

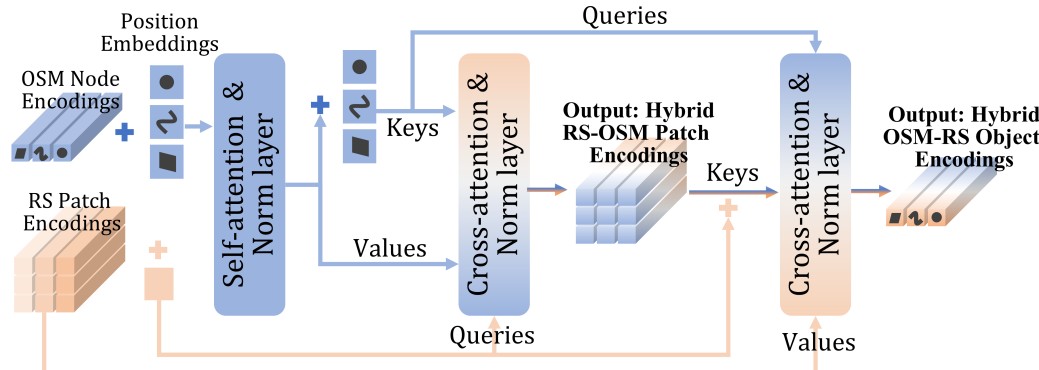

Figure 5: The detail structure of object-patch fusion encoder

As illustrated in Fig. 5, the object-patch fusion encoder is designed to effectively align and integrate RS image patches with geographic objects (node features) from OSM. It comprises a self-attention layer followed by two cross-attention layers, forming a lightweight yet expressive architecture for cross-modal fusion. The initial self-attention layer captures intra-modal relationships within the OSM object features, allowing the model to establish a coherent representation of the geographic context before engaging in cross-modal interactions.

To enhance the model's spatial sensitivity, position embeddings are added to both the keys and values prior to each attention operation. This spatial conditioning enables the encoder to better model the relative locations of RS patches and OSM objects, ensuring that attention weights are not only content-driven but also spatially aware. As a result, the model can form stronger associations between elements that are geographically close, leading to more accurate and meaningful fusion. The two cross-attention layers are asymmetrically structured to support bidirectional information exchange between modalities. The first cross-attention layer projects RS image patches as queries and OSM objects as keys/values, producing hybrid RS–OSM patch encodings that embed geographic context into visual features. Conversely, the second cross-attention layer uses OSM objects as queries to attend over RS patches, generating hybrid OSM–RS object encodings that incorporate visual cues into semantic representations of geographic objects. These two outputs are both fine-grained multimodal representations— the image patch level and the geographic object level, respectively — thereby equipping the model with the flexibility to support a broad range of downstream tasks.

## B   Details of the pretraining dataset

The pretraining dataset is derived from SkyScript-top30 [31], which contains 1271431 RS-OSM sample pairs. According to SkyScript, the RS images are originally downloaded from Google Earth Engine (GEE) platform, including 10 image collections like National Agriculture Imagery Program and Harmonized Sentinel-2, with a geographic coverage for all continents except Antarctic. The detailed information regarding the sources and distribution of RS images can be found in [31]. As for the OSM data, each RS image used in this study contains metadata on geographic location and timestamp, and we leverage them to retrieve corresponding OSM data via the Overpass API, which allows querying OSM data within a specified time range. After retrieving OSM data via Overpass, we apply rule-based cleaning to remove common issues—such as fixing or removing invalid polygons (e.g., self-intersections), eliminating duplicate/conflicting objects, and filtering tags with formatting or spelling errors.

To ensure diversity and reduce training burden, we remove sample pairs with overlapping spatial coverage and filter out samples with unavailable OSM data. According to the statistics shown in Fig. 6(a), each sample contains around ten OSM vector objects of points, polylines, and polygons on

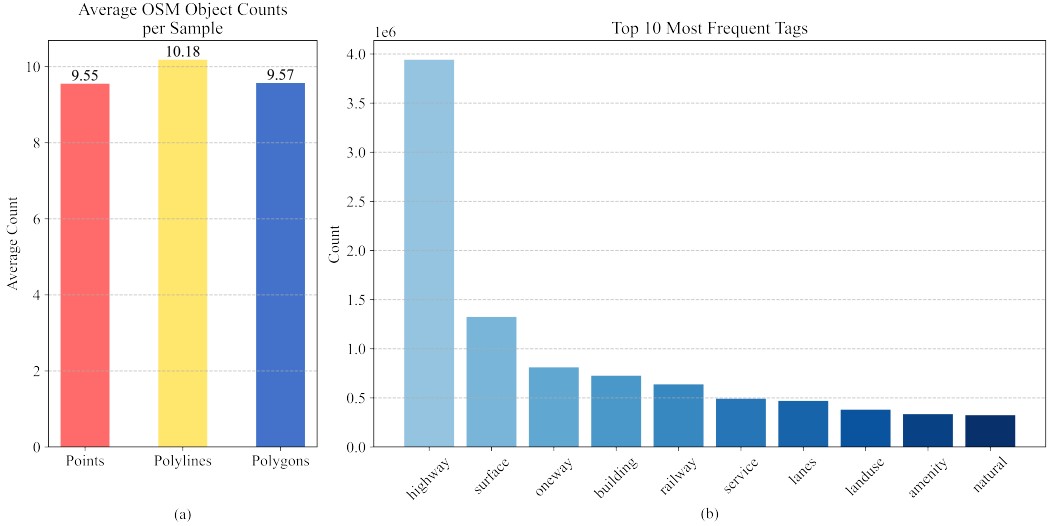

Figure 6: (a) The average counts of each kind of OSM object per sample. (b) The top 10 most frequent tags in the GeoLink pretraining dataset.

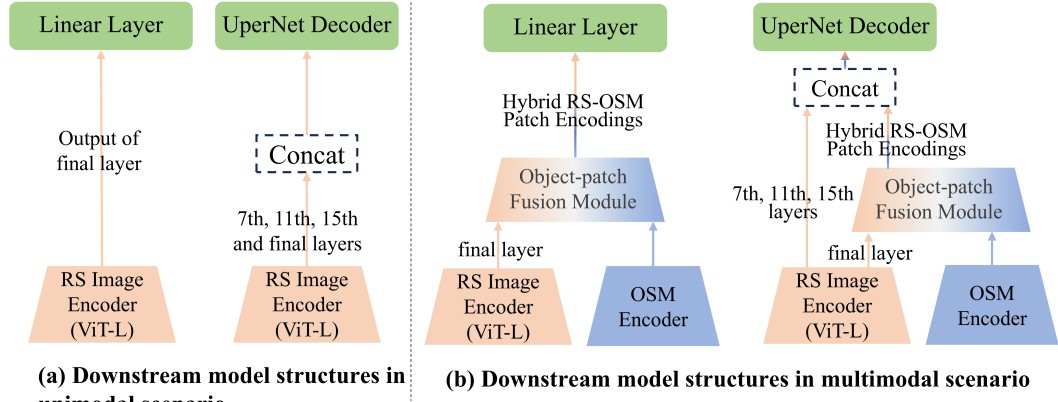

(a) Downstream model structures in unimodal scenario

(b) Downstream model structures in multimodal scenario

Figure 7: Model structures for diverse downstream tasks in unimodal and multimodal scenarios.

average. The pretraining dataset includes a total of 2,790 types of tags, with the top 10 most frequent ones shown in Fig. 6(b), namely: highway, surface, one way, building, railway, service, lanes, land use, amenity, and natural.

## C   Details of downstream task settings

### C.1   The selection of baseline models

Ideally, comparisons should be made among models with identical spectral and modality settings, but in practice, this is often difficult to achieve. The diversity of spectral configurations in current RS FMs indeed makes idealized comparisons challenging, as many models are tailored to specific satellite spectral bands. For example, CROMA is built on Sentinel-1/2 data, while Prithvi-EO-2.0 [1] uses six channels shared by Sentinel-2 and Landsat: Blue, Green, Red, NIR, SWIR1, and SWIR2. Therefore, several prior works [1, 17, 18] have adopted a pragmatic approach by evaluating models with different spectral inputs on the same benchmark datasets. Although this may not ensure fully aligned comparisons, it provides meaningful insights into the models' generalization on common datasets.

To the best of our knowledge, there is currently no RS FM that uses the exact same data types as GeoLink (RS and OSM data), making it difficult to directly compare with fully modality-aligned baselines. The starting point of our experimental design is to explore how, and to what extent,

Table 8: Detailed information of benchmark datasets for multimodal downstream tasks: UFZ, UV, POP, and CO2.

| Benchmark | UFZ | UV | POP | CO2 |
|---|---|---|---|---|
| Experimental region | Chicago metropolitan area, Singapore, Shenzhen | Beijing, Shanghai | Same as UFZ | Same as UFZ |
| RS image source | ArcGIS World Imagery, Bing Map | ArcGIS World Imagery | ArcGIS World Imagery, Bing Map | ArcGIS World Imagery, Bing Map |
| GSD | 1m, 3m | 1m | 1m, 3m | 1m, 3m |
| Bands | RGB | RGB | RGB | RGB |
| Annotation source | Official statistics: Chicago Singapore Shenzhen | Expert annotation | WorldPop Dataset | ODIAC Fossil Fuel Emission Dataset |
| Annotation processing | Manual refinement, reclassification, spatially aligned cropping | Spatially aligned cropping | Resampling, spatially aligned cropping | Resampling, spatially aligned cropping |
| Image cropping size | 224×224 | 224×224 | 224×224 | 224×224 |
| Sample count | 60,970 | 1,899 | 59,284 | 47,607 |

integrating OSM data can enhance RS FM. To fairly and comprehensively evaluate GeoLink, we consider baselines designed from different perspectives: (1) Unimodal baselines that use the same RS spectral modality as GeoLink, including GASSL, ScaleMAE, and Cross-scale MAE; (2) Multimodal baselines that incorporate additional modalities, such as MMEarth, CROMA, and DOFA. Different strategies are employed to adapt multimodal baselines to the RGB-only testing benchmark. The original DOFA model includes a dynamic projection module as a tokenizer to normalize various spectral inputs, allowing it to directly process RGB images without additional modifications. For CROMA and MMEarth, we follow PANGAEA-bench [17] employs zero-padding to fill in missing spectral bands for input.

## C.2 Model structures for downstream task evaluation

As illustrated in Figure 3, we present the model architectures employed for various downstream tasks under both unimodal and multimodal settings. For linear classification and regression tasks, the unimodal approach directly utilizes the encodings output from the final layer of the RS image encoder—either the [cls] token or the mean of patch features—as input to a linear layer (left side of Fig. 7(a)). In the multimodal setting, this input is replaced by the mean of the hybrid RS-OSM patch encodings (left side of Fig. 7(a)). For semantic segmentation, we adopt the UperNet decoder. Under the unimodal setting, encodings from the 7th, 11th, 15th, and final layers of the RS encoder are fed into the decoder. In the multimodal setting, the features from the final layer are replaced with the hybrid RS-OSM patch encodings, while all other components remain consistent with the unimodal configuration. Overall, aside from minor differences in feature dimensions, the model architecture remains consistent between the unimodal and multimodal settings. The multimodal encoder can function as a plug-and-play component that integrates seamlessly into the RS FMs, and we combine it with Scale-MAE to build Scale-MAE+OSM for evaluation in Sec. 4.

## C.3 Construction of the multimodal benchmark datasets

In this study, we construct three multimodal benchmark datasets to evaluate GeoLink's capability in addressing complex and comprehensive geographic tasks, including urban functional zone segmentation (UFZ), urban village identification (UV), population density estimation (POP), and carbon emission estimation (CO2). Each dataset consists of spatially aligned RS images, OSM data, and the corresponding task-specific annotations. Table 8 is a detailed overview of the construction procedures and key specifications of the four datasets. To begin with, considering factors such as geographic

Table 9: Hyperparameters for downstream tasks. LP: linear probing, Frz: freezing, FT: finetune, Cls: classification, Seg: segmentation, CD: change detection, LR: learning rate, CE: cross-entropy, MSE: mean square error

| Task | Cls (LP) | Cls (FT) | Seg/CD (Frz) | Seg/CD (FT) | UFZ/UV | POP/CO2 |
|------|----------|----------|--------------|-------------|--------|---------|
| Optimizer | AdamW | AdamW | AdamW | AdamW | AdamW | AdamW |
| Batch size | 64 | 64 | 64 | 64 | 32 | 256 |
| LR | $\{1,3,5\}$ | $\{1,3,5\}$ | 1e-4 | 1e-4 | $\{1,3,5\}$e | $\{1,3,5\}$e |
| | e$\{-2,-3,-4\}$ | e$\{-2,-3,-4\}$ | | | $\{-3,-4,-5\}$ | $\{-1,-2,-3\}$ |
| LR multiplier | – | 0.01 | – | 0.01 | 0.01 | 0.01 |
| Weight decay | 0.05 | 0.05 | 0.05 | 0.05 | 0.05 | 0.05 |
| Beta | 0.9, 0.999 | 0.9, 0.999 | 0.9, 0.999 | 0.9, 0.999 | 0.9, 0.999 | 0.9, 0.999 |
| Epoch | 50 | 50 | 80 | 80 | 30 | 30 |
| LR scheduler | Multi-step | Multi-step | Multi-step | Multi-step | Cosine | Cosine |
| Default splits | 50/10/40 | 50/10/40 | Same as | Same as | 50/10/40 | 50/10/40 |
| (train/val/test) | % | % | PANGAEA-bench | PANGAEA-bench | % | % |
| Loss function | CE | CE | CE/dice | CE/dice | CE | MSE |

characteristics, level of development, and data availability, we select several representative experimental regions for the four tasks, including the Chicago metropolitan area, Singapore, Shenzhen, China and so on. High-resolution RS images for these regions are acquired from ArcGIS World Imagery and Bing Map, covering three RGB bands with GSDs of 1m or 3m. Corresponding OSM data are retrieved via the Overpass API. Subsequently, we construct high-quality annotated labels required for the four tasks through a combination of web data collection, expert annotation, and manual refinement. For the Urban Functional Zone (UFZ) task, we obtained original urban planning data from official statistics (sources listed in the table below), which are further refined using auxiliary references such as Google Maps to ensure their reliability. These refined data were then reclassified into the nine-category UFZ taxonomy: water, green space, farmland, undeveloped land, residential, commercial, institutional, industrial, and transportation. For the UV task, labels were generated entirely through manual annotation. For the POP and CO2 tasks, we leverage two well-established reanalysis datasets to obtain annotations—WorldPop and the ODIAC Fossil Fuel Emission Dataset—both of which have been extensively validated in prior research [66, 67, 68, 69, 70]. We process the RS images, OSM data, and annotations through resampling, reclassification, and other methods to obtain the final benchmark datasets.

## C.4 Details of downstream evaluation protocols

To ensure the reproducibility of our experiments, we provide detailed hyperparameter settings for all downstream tasks in Table 9. All tasks are optimized using AdamW with a weight decay of 0.05 and $\beta$ values set to [0.9, 0.999]. To preserve the original characteristics of the data, no data augmentation is applied in any of the tasks. For classification tasks, we perform a grid search over learning rates and report the best results for each model on each dataset. The data is split into 50% for training, 10% for validation, and 40% for testing. All results are averaged over three runs with different random seeds. For semantic segmentation and change detection tasks, all the settings of data split, learning rate, and loss function follow the default of the PANGAEA-bench [17]. Notably, at the time of paper submission, PANGAEA-bench has not yet released an officially processed version of the FiveBillionPixel dataset. Therefore, we use the dataset provided in the original FiveBillionPixel paper [58], crop the images to a size of $520 \times 520$ as input, and follow the original data split for consistency. For the UFZ, UV, POP, and CO2 tasks, we also conduct learning rate searches individually to ensure optimal performance for each task.

# D Other experiments

## D.1 Performance under limited annotations

Performing well under limited labeled data is one of the key metrics for evaluating FMs. To assess this, we conduct experiments using only 10% of the samples on both linear probing and UFZ tasks. Table 10 reports the average results of linear probing across seven datasets, as well as the results

for UFZ under unimodal and multimodal settings. Compared with Fig. 1 and Fig. 3, GeoLink demonstrates even more significant advantages in this scenario, indicating its stronger adaptability to downstream applications with limited training samples.

Table 10: Performance comparison on linear probing (top-1 accuracy %) and UFZ (mIoU %) tasks using only 10% samples for training.

| Model | Linear probing | UFZ-unimodal | UFZ-multimodal |
|---|---|---|---|
| Scale-MAE | 83.04 | 39.08 | – |
| GeoLink | 86.17 | 48.48 | 53.12 |

## D.2 Impact of loss weights

GeoLink incorporates three distinct learning objectives, and their relative weighting can influence the model's performance. In this study, we always fix the weight of the image reconstruction loss to 1 and focus on adjusting the weights of the region-image contrastive loss and the spatial consistency loss. The default weight for both is set to 0.01, and we conduct experiments by scaling each of them up and down by an order of magnitude. The results are presented in Table 11. We evaluate performance on two downstream tasks: linear probing and UFZ-multimodal. The results reveal a differentiated impact of the two losses. The region-image contrastive loss primarily affects the performance of linear probing, indicating its dominant role in optimizing the image encoder. In contrast, variations in the spatial consistency loss have a greater influence on the UFZ-multimodal task, suggesting it plays a crucial role in enhancing cross-modal feature fusion. Setting both loss weights around 0.01 yields a favorable balance, facilitating effective synergy among the three objectives.

Table 11: Ablation study of contrastive and consistency loss weights. Evaluation includes linear probing (top-1 accuracy %) and UFZ-multimodal (mIoU %).

| Contrastive loss | Consistency loss | Linear probing | UFZ-multimodal |
|---|---|---|---|
| 0.01 | 0.01 | 92.60 | 60.00 |
| 0.1 | 0.01 | 91.37 | 58.43 |
| 0.001 | 0.01 | 91.89 | 58.74 |
| 0.01 | 0.1 | 91.56 | 57.98 |
| 0.01 | 0.001 | 92.31 | 56.18 |

## D.3 Key-point number for position embedding

Table 12: Ablation study on the number of key-points for polyline and polygon representations.

| Polyline key-points | Polygon key-points | Linear probing | UFZ-multimodal |
|---|---|---|---|
| Centroid + endpoints | Centroid + 3 sampling points | 92.60 | 60.00 |
| Centroid | Centroid | 92.32 | 59.46 |
| Centroid + endpoints | Centroid + 5 sampling points | 91.77 | 58.02 |

Since sinusoidal position embedding is not inherently designed to represent the spatial characteristics of polyline and polygon vectors, we propose to approximate their spatial coverage by sampling a set of representative key-points. The number of sampled points can directly influence the spatial representation and, consequently, the model's performance. Therefore, in this section, we conduct experiments to explore the impact of key-point number on downstream tasks. First, we represent the positions of both polyline and polygon solely by their centroids, without sampling any key-points. The results of this setting are shown in the second row of the Table 12. Interestingly, using only the centroid as the position proxy still yields competitive performance, with only a slight drop compared to the default setting (first line). Next, we fix the polyline key-points while increasing the number of polygon key-points. Specifically, using the centroid as the center, we randomly sample five (the defalt is three) additional points within the polygon at varying radii and angles, forming a total of

six points to represent its spatial extent. The corresponding results are presented in the third row of the Table 12, where we observe a noticeable performance degradation. This can be attributed to the nature of sinusoidal position embeddings: they encode positional information through a combination of multi-frequency sine and cosine functions. Averaging multiple such embeddings tends to smooth out high-frequency variations, potentially over-smoothing the positional signal and impairing the model's ability to distinguish spatial patterns. In future work, we aim to explore more expressive and principled methods of position encoding that can seamlessly handle point, polyline, and polygon geometries within a unified framework, thereby enabling more effective spatial correlation.

## D.4 Visulization of GeoLink mapping results in UFZ and UV tasks

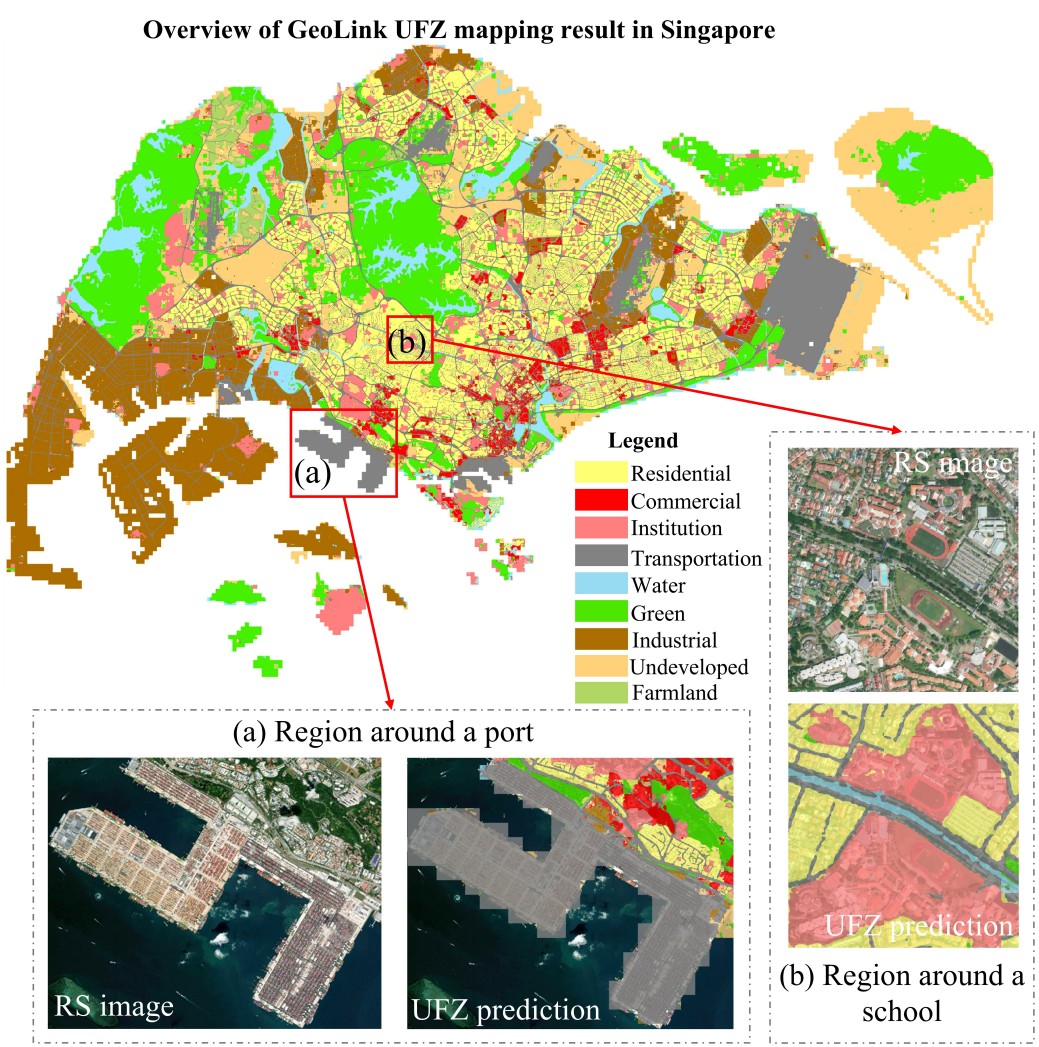

Figure 8: Overview and details of GeoLink UFZ mapping result in Singapore.

Large-scale geographic mapping is one of the most important real-world applications of RS data, and also a core capability that RS FMs should possess. In this section, we move beyond evaluating GeoLink solely through quantitative metrics, and instead demonstrate its potential for regional geographic mapping by visually comparing its predictions with real-world spatial layouts. Specifically, we leverage GeoLink to perform full-coverage predictions for Singapore and Beijing (multimodal setting) on UFZ segmentation and UV identification tasks, respectively. The predicted results are stitched together based on the geo-coordinates to generate complete UFZ map for Singapore and UV map for Beijing. As shown in Fig. 8, we first present the overall UFZ mapping results for Singapore. It is evident that the predicted spatial layout aligns well with the actual urban structure of

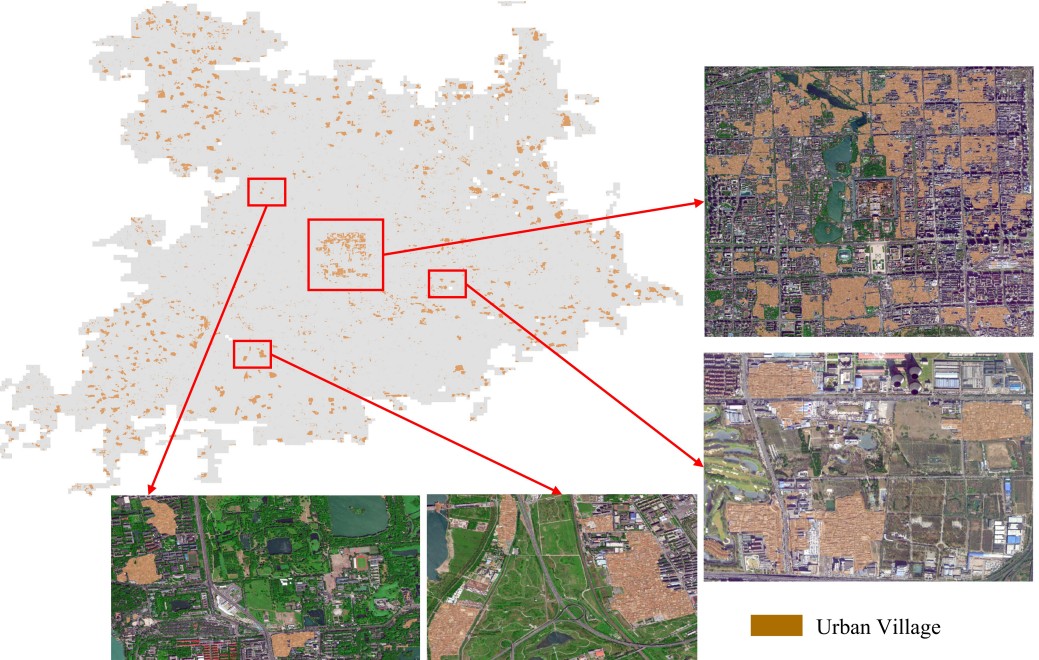

**GeoLink UV mapping result in Beijing**

Urban Village

Figure 9: Overview and details of GeoLink UV mapping result in Beijing.

Singapore. For instance, the airport in the upper right and the industrial zone in the lower left are both accurately delineated. Even in the central area, where residential, commercial, and institution zones are intermingled, GeoLink is able to make reasonably precise distinctions. We further illustrate two representative local regions in detail: (a) the area surrounding the port (transport zone), where GeoLink clearly captures the spatial boundaries of the port infrastructure; (b) the area around a school (institution zone), where the model effectively leverages semantic information from OSM data and road boundary texture from RS image to delineate the functional area, despite its complexity. Fig. 9 showcases the results of urban village extraction and visualization in the Beijing region. UVs in Beijing are primarily concentrated within the second ring road, and GeoLink successfully identifies both the location and spatial extent of them (upper right), demonstrating its effectiveness in this task. Integrating RS FMs with mapping-related technologies to enable automated, high-precision geographic mapping holds significant promise across various geoscientific domains. At the same time, it represents a pressing technical challenge that remains to be fully addressed. We argue that future evaluations of RS FMs should incorporate real-world mapping performance as a critical metric, thereby continuously enhancing their practical applicability and deployment value in real geospatial scenarios.

### D.5 Comparison with domain generalization model

We also compare GeoLink with CrossEarth [71], a vision FM for domain generalizable RS semantic segmentation, to evaluate the generalization ability. CrossEarth provides multiple domain generalization scenarios (e.g., urban-to-rural, different climate zones) and releases models trained on various source datasets. Based on our experimental setup and the available open-source models, we select the versions trained on ISPRS Potsdam (RGB) and CASID (Temperate Monsoon), as both are closely related to land cover tasks, similar to our two benchmarks on semantic segmentation task: FiveBillionPixels and AI4Smallfarms. This setting helps reduce the impact of domain gaps and better reflect the true performance of CrossEarth. In CrossEarth's original zero-shot setting, the source and target domains share the same label taxonomy. However, this is not the case for our benchmarks. Therefore, we modify the classification head of CrossEarth to match the number of classes in our benchmarks, while keeping all other layers and pretrained weights fixed. We finetune

the classification head on the target datasets and compare the results to GeoLink with freezing encoder. This setting helps CrossEarth adapt to our benchmarks and preserve its knowledge learned from the source domain.

Experimental results in the Table 13 show that CrossEarth exhibits strong generalization ability and performs reasonably well, although slightly below GeoLink. Notably, the model pretrained on CASID outperforms the one pretrained on Potsdam, possibly due to a closer distributional similarity to our benchmark datasets. This comparison highlights both CrossEarth's strength as a domain-generalized RS segmentation FM and GeoLink's effectiveness in leveraging OSM data to enhance the generalization of RS FM.

Table 13: Comparison GeoLink with CrossEarth on semantic segmentation tasks (mIoU %)

| Model | Source domain | FiveBillionPixels | AI4Smallfarms |
|---|---|---|---|
| CrossEarth | ISPRS Potsdam | 54.33 | 38.24 |
| CrossEarth | CASID | 56.18 | 41.07 |
| GeoLink | / | 60.49 | 43.26 |

## D.6 Comparison with task-specific models

Currently, most geospatial mapping applications still employ task-specific models. To address the need for deciding between using task-specific models or FMs, we have designed an geographic transfer-ability experiment for discussion. We uitlize two task-specific models for comparison. (1) U-Net+OSM: using ImageNet-pretrained ResNet50 as the backbone to represent convolutional models. Since the fusion module in GeoLink is plug-and-play, we use the same OSM encoder and fusion encoder from GeoLink to ensure a fair comparison. (2) UpperNet+OSM: using ImageNet-pretrained ViT-L as the backbone, representing Transformer-based architectures.

Our UFZ benchmark includes cities with highly distinct geographic conditions: Singapore, Chicago (US), and Shenzhen (China). The original paper uses a mixed training/testing setting. To evaluate transfer-ability, we train models on one city and test on another. Specifically, we train GeoLink on Singapore and test on Shenzhen and Chicago. As shown in the Table14, although all models experience performance drops due to domain shift, GeoLink consistently outperform the task-specific models in the target cities, highlighting its superior generalization across regions. This demonstrates the effectiveness of pretrained FMs in capturing transferable geographic features.

Table 14: Comparison of geographic transfer-ability

| Source Region | Target Region | U-Net+OSM | UpperNet+OSM | GeoLink (multimodal) |
|---|---|---|---|---|
| Singapore | Singapore | 54.39 | 53.80 | 61.82 |
| Singapore | Shenzhen | 32.51 | 32.21 | 37.25 |
| Singapore | Chicago | 39.62 | 40.06 | 49.12 |

