# OpenReview forum: "GeoLink: Empowering Remote Sensing Foundation Model with OpenStreetMap Data"
_NeurIPS.cc/2025/Conference — NeurIPS 2025 poster_

### Official Review · Reviewer_BhhX · 2025-06-20

**Clarity:** 3
**Significance:** 2
**Originality:** 2
**Rating:** 4
**Confidence:** 4

**Summary:**

This paper presents a new self-supervised learning (SSL) algorithm, called GeoLink, for pre-training remote sensing (RS) models. Specifically, GeoLink fuses RGB satellite imagery with OpenStreetMap (OSM) data.

The pre-training algorithm consists of:
- Separately masking and encoding satellite imagery and OSM data. The OSM encoder is a Graph Neural Network (GNN), the image encoder is a ViT. The GNN-node features are computed using a language model.
- A contrastive loss term is applied between visible encodings across these two modalities
- A reconstruction loss computed using an MAE-style decoder that predicts the hidden RGB patches (without OSM fusion).
- Another reconstruction loss, called a “spatial consistency loss”, computed using an “object-patch” fusion encoder that receives masked node embeddings and predicts the initial / hidden node embeddings, while cross-attending to the visible RGB patch encodings.

The paper demonstrates that GeoLink outperforms prior pre-trained RS models on many classification and segmentation tasks.

**Questions:**

How are the multispectral baseline methods used when processing RGB-only benchmark data?

**Ethical Concerns:**

["NO or VERY MINOR ethics concerns only"]

**Final Justification:**

The authors have addressed my major concerns and thus I recommend the paper is accepted.

**Limitations:**

Yes

**Quality:**

2

**Strengths And Weaknesses:**

Strengths:
- The paper is dense and well-written
- I agree that pre-trained RS models have not yet fully leveraged data like OSM and this is a promising research direction

Weaknesses:
- The satellite images are RGB-only. In this field (pre-trained RS models), most papers use multispectral satellite images as these additional bands are useful for many downstream tasks. Furthermore, this RGB-only choice makes comparisons a bit tricky since baselines were pre-trained on multispectral data (like CROMA).
- The main contribution of the paper (to a conference like NeurIPS) is the SSL algorithm, which I have concerns about.
  - I understand that OSM can be encoded with a GNN, but can’t it also be encoded with a transformer? The field has settled on transformers for multimodal RS, so this deviation should be better motivated and empirically demonstrated (e.g., by showing it works better than a transformer encoder).
  - A language model (BERT) is used to encode OSM nodes that are fed to the GNN. I don’t see an ablation that entirely removes the GNN and just uses the BERT encodings. It is certainly possible that all the gains of GeoLink can be explained by this natural language supervision — rather than the algorithm itself.

---

> ### Author Rebuttal · Authors · 2025-07-30
>
> We are very grateful for your thoughtful comments, recognition of the clarity of our writing and promising research direction, and suggestions for improvement on image spectral and ablation study. Your suggestions greatly help us improve our analysis of GeoLink. Please find our point-by-point responses below.
> ### **Q1. The use of multispectral satellite image**
>
> Thanks for pointing this out. We agree that the inclusion of multispectral images could enhance the model. However, we would like to emphasize that the focus of this paper is to explore how to leverage the rich geographic information of OSM data in RS FM, not to include as much data as possible. Due to the challenges in OSM vector encoding and large domain gap, the use of OSM vector data in RS FMs is under-explored. **This paper aims to bridge this gap and demonstrate the value of OSM in improving the performance and task diversity of RS FM, not the designing of multispectral RS encoder.**
>
> Moreover, **GeoLink can be readily extended to support multispectral RS images by modifying the image encoder.** In its current design, GeoLink's fusion mechanism imposes no specific constraints on the image encoder. Therefore, in principle, we can replace the existing ViT-based encoder with any encoder capable of handling multispectral data, without modifying other key components of the architecture. Previous multispectral pretraining approaches—using either separate encoders (e.g., SkySense) or joint tokenization (e.g., DOFA)—offer promising ways for adapting GeoLink to multispectral data.
>
> **As mentioned in the conclusion section (Lines 345-347), making GeoLink to support multispectral inputs is an important direction**, and we are excited to explore this further in future work.
>
> ### **Q2. Comparison with multispectral pretrained models**
>
> Thank you for raising this concern. To the best of our knowledge, **there is currently no RS FM that uses the exact same data modalities as GeoLink**, making it difficult to directly compare with fully modality-aligned baselines. The starting point of our experimental design is to explore how, and to what extent, integrating OSM data can enhance RS FM. In the experiments, we are not only considering multispectral baselines but also RGB-only baselines. The comparison with baselines designed from different perspectives provides more comprehensive evluations.
>
> Moreover, **multimodal pretraining models like CROMA and DOFA are specifically designed to accommodate multispectral RS images, and their pretraining data includes RGB bands**. Therefore, their pretrained models should—and must—be capable of processing RGB RS images. Additionally, many benchmark studies for RS FMs (e.g., \[1, 17, 18\]) also compare different spectrally pretrained RS FMs on the same downstream tasks. For example, \[17\] compares CROMA (multispectral) with Scale-MAE (RGB) on RGB-only benchmarks like xView2 and \[1\] also compares Prithvi-eo-2.0 (multispectral) and Scale-MAE on RGB-only benchmarks like pv4ger. Thus, we believe it is fair to compare GeoLink with multispectral pretraining models like CROMA and DOFA.
>
> ### **Q3. The setup of multispectral baselines for RGB-only benchmarks**
>
> Thank you for your question. Regarding the testing setup, we follw Pangaea-bench \[17\] to evaluate multispectral baseline methods on our benchmark data.
>
> Specifically, due to the distinct characteristics of DOFA and CROMA, different strategies are employed to adapt them to the RGB-only testing benchmark. The original DOFA model includes a dynamic projection module as a tokenizer to normalize various spectral inputs, allowing it to directly process RGB images without additional modifications. For CROMA, we use its multispectral optical encoder (pretrained on Sentinel-2 L2A) and employs zero-padding to fill in missing spectral bands for input.
>
>
> ### **Q4. The designing of Transformer-based OSM Encoder**
>
> Thank you very much for your suggestion. As a type of vector data, the structure of OSM makes it difficult to directly process using Transformers in the same way as images. Due to their strengths in spatial modeling (e.g., effectively capturing spatial relationships), GNNs have long dominated research on encoding vector data like OSM.
>
> To leverage the power of attention mechanisms, this study does not use the original GCN but instead employed GAT to construct the OSM encoder. To further validate your suggestion, we design two variant experiments to illustrate the effectiveness of the current OSM encoder. The experimental setups are as follows:
>
> **GCN Variant**: Replace the GAT in the OSM encoder with GCN while keeping all other settings unchanged.
>
> **Transformer Variant**: Replace the existing GNN-based OSM encoder with a Transformer containing two attention layers. To adapt to the Transformer’s structure, each OSM node in the original GNN is treated as a token input to the Transformer. Additionally, position embedding is added to each token to preserve spatial information. All other settings remain unchanged.
>
> The experimental results are shown in the table below:
>
> |OSM Encoder Setting|kNN|Linear|UFZ-U|UFZ-M|
> |-|-|-|-|-|
> |Default|87.06|92.60|55.40|60.00|
> |GCN| 87.01|92.14|55.02|59.42|
> |Transformer|86.28|91.34|53.99|58.73|
>
> Compared to the GCN variant, the attention mechanism introduced by GAT yields better performance. The Transformer variant underperforms both GCN and GAT, which we speculate is primarily due to the insufficient message-passing between OSM nodes in it, hindering the learning of OSM spatial consistency objective during pretraining.
>
> In summary, **the experimental results demonstrate that the current OSM encoder works synergistically with other components of GeoLink and achieves satisfactory performance.** In the future, we will explore more sophisticated Transformer-based OSM encoder designs to pursue potential additional improvements.
> ### **Q5. Ablation studies by removing GNN**
>
> Thank you very much for proposing such a valuable ablation study design! In fact, **the gains of GeoLink stem from the synergy of multiple components in the overall architecture, including the semantic and geometric information from OSM, as well as the structural design and learning objectives that effectively transmit this information to model training.**
>
> Following your suggestion, we design an ablation experiment: removing the OSM encoder, using a frozen BERT to encode OSM tags, applying average pooling to aggregate tag encodings of multiple OSM objects within a sample, and then using a linear projection layer to align the features with image encodings for contrastive learning. Notably, this ablation approach also renders GeoLink's original spatial consistency loss and object-patch fusion encoder unusable. Thus, the pretraining signals for the model come solely from image reconstruction and cross-modal contrastive loss, achieving what you refer to as "natural language supervision". The experimental results are shown below:
>
> |OSM Encoder Setting|kNN|Linear|UFZ-U|UFZ-M|
> |-|-|-|-|-|
> |Default|87.06|92.60|55.40|60.00|
> |No GNN|85.01|90.37|52.66|/|
>
> Compared to the original GeoLink, the ablated setup exhibits a significant performance drop, indicating that the gains of GeoLink do not come solely from natural language supervision. According to GeoLink's design, a key aspect of OSM's supervisory signal lies in using vectors' spatial information to guide the tags' semantic information for fine-grained interaction and fusion. The implementation of this learning signal requires the synergy of the OSM encoder, fusion encoder, and learning objectives. Removing the GNN-based OSM encoder disrupts this mechanism, making it difficult to achieve comparable results.
>
> Therefore, the gains of GeoLink arise from the synergistic design of multiple components in the architecture, not just natural language supervision.

---

> > ### Comment · Reviewer_BhhX · 2025-08-05
> >
> > I thank the authors for the rebuttal. They have addressed my two major concerns (GNN vs. transformer and language supervision) with new experiments, thus I will increase my score.
> >
> > > GeoLink can be readily extended to support multispectral RS images by modifying the image encoder.
> >
> > Yes, but it's another thing to demonstrate it. Many SOTA RS FMs are pre-trained on multispectral inputs. As a result, GeoLink is missing comparisons to them. Zero-padding inputs of such models, like CROMA, is not ideal despite it being done in other papers (not CROMA itself).
> >
> > > Replace the existing GNN-based OSM encoder with a Transformer containing two attention layers.
> >
> > I assume the authors mean two _transformer_ layers/blocks, not just two self-attention layers — since FFNs should also be included. This is a small transformer. What is the FLOPs of this small transformer and how does it compare with the GNN used in GeoLink? Maybe the GNN wins because it is larger?

---

> ### Author Response · Authors · 2025-08-07
> **Thank you for your comment**
>
> We sincerely thank the reviewer for taking time to re-evaluate our work. Your insightful suggestions regarding the GNN vs. Transformer comparison and the role of language supervision helped us clarify key aspects of our method. We truly appreciate your recognition of our additional experiments and your willingness to raise the score.
>
> ### **Q1. Comparison with CROMA**
> Thank you very much for your comment. We completely understand your concern. The diversity of spectral configurations in current RS FMs indeed makes idealized comparisons challenging, as many models are tailored to specific satellite spectral bands. For example, CROMA is built on Sentinel-1/2 data, while Prithvi-EO-2.0 uses six channels shared by Sentinel-2 and Landsat: Blue, Green, Red, NIR, SWIR1, and SWIR2. Ideally, comparisons should be made among models with identical spectral and modality settings, but in practice, this is often difficult to achieve. Therefore, several prior works have adopted a pragmatic approach by evaluating models with different spectral inputs on the same benchmark datasets. Although this may not ensure fully aligned comparisons, it still provides meaningful insights into the models' generalization on common datasets. In our study, we selected a wide range of publicly available RS FMs as baselines to ensure a comprehensive comparison, including RGB-only models (GASSL, Scale-MAE, Cross-scale MAE), a spectrally adaptive baseline (DOFA), and multispectral baselines (CROMA and the newly added MMEarth). **In the revised paper, we will also add the discussion on RS FM evaluation practices to reflect these complexities.**
>
> While GeoLink has demonstrated the value of OSM data in enhancing both the performance and task diversity of RS FMs, we plan to focus on designing spectrally adaptive RS encoders to further extend its multispectral capabilities. Thank you for raising this point and for your thoughtful engagement with our work.
>
> ### **Q2. The designing of Transformer-based OSM Encoder**
> Thank you for your insightful comment. In the OSM encoder variant experiments, **the inferior performance of the Transformer variant compared to the default GNN-based OSM encoder is not due to a smaller model size. On the contrary, the Transformer variant actually has more parameters than the default encoder.**
>
> To keep the model sizes comparable, the Transformer variant used in our rebuttal adopts a two-layer MultiHeadAttention architecture. In light of your suggestion that FFNs should also be included, we design a new variant using the standard Transformer block, which we refer to as the S-Transformer variant. The table below summarizes the main architectural components, parameter counts, and FLOPs of the default GNN-based OSM encoder, the Transformer variant, and the S-Transformer variant. For FLOPs estimation, we assume an input sample with 120 nodes (40 for each of the three node types) and 450 edges (50 for each of the nine edge types, used only in the GNN-based encoder). The table shows that both variants have larger parameter counts and FLOPs than the default OSM encoder.
>
> |OSM encoder setting|Main structure|Total parameters|FLOPs|
> |-|-|-|-|
> |Default|9 GATConv Layers (in_dim=512, out_dim=128, Fig. 5)|593280|94.79 M|
> |Transformer|Linear(512→256), 2× MultiHeadAttention (embed_dim=256, heads=8), Linear(256→128) |692096|196.16 M|
> |S-Transformer|Linear(512→256), 2× Transformer blocks (embed_dim=256, ffn_dim=1024, heads=8), Linear(256→128)|1743744|441.48 M|
>
> The experimental results are shown in the following table. Across four tasks, replacing two attention layers with two standard Transformer blocks (which introduces more parameters) does not lead to significant performance changes. Moreover, both variants perform noticeably worse than the default GNN-based encoder. We believe this is primarily due to the Transformer's limitations in modeling spatial relationships between OSM nodes, which impairs effective message passing and ultimately weakens GeoLink's pretraining effectiveness. This observation aligns with prior research, where GNNs remain the dominant approach for encoding vector-based geospatial data \[\*1\]. We also find your suggestion of unifying OSM and RS encoders via Transformers very insightful, as it may contribute to building a more unified framework for GeoAI in the future.
>
> |OSM Encoder Setting|kNN|Linear|UFZ-U|UFZ-M|
> |-|-|-|-|-|
> |Default|87.06|92.60|55.40|60.00|
> |Transformer|86.28|91.34|53.99|58.73|
> |S-Transformer|86.14|91.47|53.73|58.57|
>
> **We will add these OSM variant experiments and the additional ablation studies on language supervision in the revised version of the paper.** And your insightful suggestions have greatly contributed to improve the clarity and completeness of our paper. We sincerely appreciate the time and effort you devoted to reviewing our paper.
>
> \[*1\] Chen Y, Huang W, Zhao K, et al. Self-supervised representation learning for geospatial objects: A survey[J]. Information Fusion, 2025: 103265.

---

> > ### Comment · Reviewer_BhhX · 2025-08-07
> >
> > Thank you for the discussion and these results. I will recommend the paper is accepted.

---

> > > ### Author Response · Authors · 2025-08-07
> > > **Thank you for your comment**
> > >
> > > We sincerely thank you for your positive feedback and recommendation for acceptance. We truly appreciate your recognition of our work!

---

### Official Review · Reviewer_6XcF · 2025-06-27

**Clarity:** 3
**Significance:** 3
**Originality:** 3
**Rating:** 3
**Confidence:** 4

**Summary:**

This paper introduces GeoLink, a multimodal framework that integrates OpenStreetMap (OSM) data with remote sensing (RS) foundation models to enhance geospatial intelligence. GeoLink leverages OSM data to enrich the RS foundation model during both pretraining and downstream task stages through multi-granularity learning signals, masked input pretraining, and RS-OSM fusion. The framework demonstrates significant improvements in the performance of the RS image encoder and shows strong adaptability to complex geographic scenarios across various tasks.

**Questions:**

1. In the multimodal downstream tasks, the hybrid RS-OSM patch encodings enrich each visual patch with OSM context. This works by attending to OSM object features. Is there a risk of spatial "blurring" where a patch at the edge of a large feature (e.g., a large "park" polygon) receives the same aggregated contextual vector as a patch in the center? How does the model differentiate fine-grained intra-object location? Does the sinusoidal positional embedding for the patch itself provide enough signal to mitigate this?
2. The ablation study on OSM completeness (Table 4) is insightful. However, it relies on random removal. In reality, OSM data quality is systematically biased (e.g., better coverage in developed urban centers than in rural areas). How might this geographic bias in pretraining data affect the model's generalization performance? Does the model perform better on downstream tasks in regions that are well-represented in the pretraining dataset?
3. The paper provides excellent ablations on the learning objectives. Could you provide some insight into the design of the OSM encoder itself? How critical are the different topological edge types? For instance, what would be the performance impact of a simpler model that only considers a "disjoint" (via Delaunay) and a generic "intersects/contains" relationship, thereby reducing the GNN's complexity?
4. The framework is designed for OSM. How do you envision GeoLink being adapted for other types of geospatial vector data that may have different semantics or structures, such as official administrative boundaries, cadastral (property) maps, or utility networks? Would the general framework of a GNN encoder + fusion module still apply?
5. The experimental comparison is insufficient. (1. GeoLangBind-L needs to be added as a comparison method. (2. DOFA needs to adopt the ViT-L version; otherwise, the comparison would be unfair.
6. Does the result of Scale-MAE+OSM indicate that the proposed designed fusion encoder is a plug-and-play module and whether it can be attached to other advanced models for experimentation?

**Ethical Concerns:**

["NO or VERY MINOR ethics concerns only"]

**Final Justification:**

I maintain my score. The author addressed my questions. But I think the technical contribution of this work is still rather limited.

**Limitations:**

yes

**Quality:**

3

**Strengths And Weaknesses:**

Strengths
1. GeoLink innovatively combines RS and OSM data into a comprehensive multimodal framework. This integration not only addresses the structural differences between the two data types but also enhances the semantic understanding of RS images through spatial context.
2. By utilizing the multi-granularity nature of OSM data, GeoLink designs region-image level alignment and object-patch level interaction to capture information across different spatial scales.
3. Adopting a masked input approach for both RS images and OSM graphs during pretraining, GeoLink achieves efficient training while maintaining performance improvements.

Weaknesses

1. The paper does not discuss the computational scalability of the GNN-based OSM encoder and the object-patch fusion module, particularly in regions with very high object density. In dense urban cores, a single scene can contain thousands of OSM objects, which could make the graph construction, message passing, and especially the cross-attention mechanism computationally prohibitive. A discussion on these limitations and potential mitigation strategies (e.g., graph sampling) would strengthen the paper.
2. While the overall framework is highly novel, the individual building blocks (ViT-MAE, GATConv, InfoNCE, cross-attention) are established techniques.

3. Although GeoLink shows some robustness to incomplete OSM data, the performance may still be affected in regions with sparse OSM coverage. The authors need to further explore methods to reduce this dependency or supplement missing data.

4. Despite faster convergence during pretraining, the multimodal framework involving multiple encoders and complex GNN structures may lead to higher computational costs during training and inference. The training is fast, but don't know how efficient it is during inference. The paper does not discuss the inference time of GeoLink, which is crucial for real-time applications. The authors should evaluate and improve the model's inference efficiency.

---

> ### Author Rebuttal · Authors · 2025-07-30
>
> We are very grateful to your thoughtful comments and recognition of GeoLink's novelty, quality, and significance. We also appreciate your concerns on the computational scalability, OSM data, and so on, which will help us improve the quality of the paper. Please find our point-by-point responses below.
>
> ### **Q1. The computational scalability**
> Thank you for your insights on the model architecture. We understand the concern, but **based on our current design, the OSM encoder and fusion module do not significantly increase the computational cost.**
>
> First, both the OSM encoder (9 GAT layers, Fig. 5) and the fusion module (3 attention layers, dim=256, Fig. 6) are lightweight, and very few training parameters are introduced.
>
> Second, during pretraining, the fusion module only applies cross-attention between masked 20% OSM nodes and unmasked 25% RS patches, greatly reducing computation. The table below shows memory usage with/without the fusion module under standard settings.
>
> |Fusion encoder|Pretraining memory|
> |-|-|
> |With|46955 MB|
> |Without|46121 MB|
>
> Third, the fusion module outputs lower-dimensional features (256 vs. 1024 in ViT-L). In downstream tasks like semantic segmentation, where decoder size depends on feature dimensions, this can reduce memory usage. As shown in the table, the multimodal setting uses less memory than the unimodal one in the UFZ segmentation task.
>
> |Mode|UFZ memory|
> |-|-|
> |Unimodal|17105 MB|
> |Multimodal|11517 MB|
>
> To address concerns about high memory usage from dense OSM regions, we currently limit the number of OSM nodes to 1000 per sample through random sampling. Our data shows that fewer than 0.01% of samples exceed this threshold, so their impact on training is minimal. In the future, we plan to explore more targeted sampling strategies based on entropy or clustering. For extremely dense cases, we can apply strategies like cropping the spatial coverage or increasing RS image resolution to reduce the number of included OSM objects.
> ### **Q2. Bias in OSM data**
> Thank you for your insightful comments and for recognizing the value of our ablation study. Geographic bias caused by the distribution of pretraining data is an objectively common issue in FMs \[19\], and GeoLink is no exception. However, **it is inherently difficult to disentangle whether the performance differences across regions stem from geographic bias encoded in the FMs, or from the intrinsic spatial heterogeneity between regions.**
>
> We conduct UFZ segmentation task separately in Shenzhen (China) and Chicago (US) to illustrate this. As noted in \[*1\], OSM coverage in China is considerably lower than in the US. Results in the table below show that GeoLink does not perform as well in Shenzhen as it does in Chicago. While this might be partially attributed to geographic bias, it is not sufficient to conclude that GeoLink exhibits significant bias, as as Shenzhen's complex urban structures make the task more challenging. As highlighted in \[19\], there is no work on accurately quantifying geographic bias in FMs currently. Designing convincing experiments to isolate geographic bias in FMs remains difficult, and we believe this is an exciting and important direction for future research.
>
> On the other hand, there are some practical ways to mitigate geographic bias during FM pretraining:
>
> **Data-side**: As you noted, OSM data quality is systematically biased, and we can mitigate this by increasing sampling density in OSM-sparse regions;
>
> **Model-side**: Developing globally location-aware FMs that treat geographic space as a continuous input to guide the modeling of prior knowledge of geographic distribution, as explored in \[42].
>
> |Region|Unimodal|Multimodal|
> |-|-|-|
> |Shenzhen|34.12|40.45|
> |Chicago|47.28|55.10|
>
> ### **Q3. Inference efficiency**
> We appreciate the reviewer's concern. In our default settings, GeoLink's inference speed on the UFZ segmentation task is 83.2 samples/sec for unimodal and 76.0 samples/sec for multimodal. The slower inference in the multimodal setting is typical due to increased data and model modules. However, this speed is acceptable for most practical applications. For time-sensitive tasks, we plan to use model distillation or data caching to improve efficiency.
>
> ### **Q4. Spatial "blurring" effect**
> Thank you for your insightful comments. In GeoLink, each image patch is assigned a distinct position embedding based on its spatial location, leading to different cross-attention interactions with the same OSM object. Therefore, **a patch at the edge of a large object receives a different aggregated contextual vector to a patch in the center, reflecting spatial distinctions within an object.**
>
> Distincting fine-grained intra-object location requires an accurate and complete representation of polygon geometry—an ongoing challenge in vector-based geospatial encoding. In this work, we adopt sinusoidal position embeddings combined with keypoint sampling to approximate polygon shapes. Empirically, the outstanding performances of GeoLink on multimodal fine-grained tasks demonstrate that this strategy is effective and fine-grained spatial interaction contributes meaningfully to data fusion. **As described in Lines 347-350, we believe that more accurate position embeddings could further enhance GeoLink**. Recent works on geo-vector encoding like Poly2Vec is particularly inspiring for improving our position embedding scheme. In the future, we hope more studies can be proposed to focus on positional modeling approaches for fine-grained multimodal geospatial data fusion.
> ### **Q5. Ablation studies for OSM encoder**
> Thank you for your suggestion and recognition of our ablation experiments. We conduct two additional ablation studies:
>
> **Less Edge**: Only five types of spatial relations (Delaunay, intersect, within, cross, and overlap) are retained;
>
> **No Edge**: All spatial edges are removed.
>
> As shown in the table, the "Less Edge" setting causes a slight performance drop, highlighting the value of diverse spatial relations. The "No Edge" variant performs worse, likely because without edges, the OSM encoder lacks message passing—masked nodes can't gather spatial context to query image features, which weakens the pretraining process.
>
> |OSM Encoder Setting|kNN|Linear|UFZ-U|UFZ-M|
> |-|-|-|-|-|
> |Default|87.06|92.60|55.40|60.00|
> |Less Edge|86.95|91.97|55.02|57.84|
> |No Edge|85.20|89.47|52.88|55.65|
>
>
> ### **Q6. Adapting GeoLink to other geospatial vector data**
> From a data structure view, **GeoLink's vector modeling strategy is applicable to other geospatial vector datasets, such as cadastral maps.** Its fusion module remains effective regardless of the specific vector kind. However, from an information content view, some geospatial data—such as administrative boundaries—contain limited fine-grained information. These datasets might be better suited as spatial constraints during training rather than as rich semantic sources, and thus may offer limited enhancement to GeoLink. **Therefore, adapting GeoLink to different types of vector data should be guided by the specific characteristics and information content of the data.**
> ### **Q7. Comparison with GeoLangBand-L and DOFA-L**
> Thank you for your suggestions. We test GeoLangBand-L and DOFA-L, and the results are as follows. GeoLink outperforms DOFA-L in all tasks and outperforms GeoLangBand-L in 4/7 tasks. It should be noted that unlike RS FMs, **GeoLangBand is a vision-language model (VLM) finetuned on SigLIP, which makes the comparison unfair.** Specifically, they are fundamentally different since:
>
> 1. Like most RS VLMs, GeoLangBand is not trained from scratch—its SigLIP backbone is pretrained on 400M image-text pairs. In contrast, RS FMs are generally trained from scratch and VLMs are not used as baselines in previous RS FM works \[3, 10, 32, 33, 34\] as well.
> 2. GeoLandBand also distills from DinoV2 (1.2B samples) and ImageNet ViT. Totally, the pretraining data accessed by GeoLink (1.27M) accounts for less than 0.08% of that in GeoLangBand (more than 1.6B).
> 3. VLMs are typically evaluated in zero-shot settings and focus on aligning images with textual semantics, which is fundamentally different from GeoLink's motivation—leveraging freely available geospatial data to improve RS FM.
>
> Thus, based on resource consumption, design motivation, and learning paradigm, GeoLangBand should be benchmarked against other CLIP/SigLIP-based RS VLMs, rather than RS FMs pretrained from scratch.
>
> |Model|Cls (kNN)|Cls (Linear)|Cls (FT)|Seg (Freezing)|Seg (FT)|CD (Freezing)|CD (FT)|
> |-|-|-|-|-|-|-|-|
> |GeoLangBand-L|88.98|93.12|94.98|51.47|55.92|61.23|62.72|
> |DOFA-L| 84.67|91.24|93.16|48.07|54.94|60.12|61.95|
> |GeoLink|87.06|92.60|94.53|51.88|56.11|61.58|63.00|
>
> ### **Q8. Plug-and-play fusion encoder**
> The fusion encoder is plug-and-play and can be used to enhance other unimodal models. Combining suggestions from you and Reviewer gYwk, we combine the fusion module with three semantic segmentation networks for the UFZ task:
>
> **U-Net**: Uses an ImageNet-pretrained ResNet50 backbone, representing CNNs.
>
> **UpperNet**: Uses an ImageNet-pretrained ViT-L backbone, representing Transformers.
>
> **MSSNet**: A multi-scale semantic segmentation network \[*2\] specifically designed for UFZ tasks.
>
> Performances of all three models are significantly improved after integrating the fusion module, verifying its plug-and-play flexibility.
>
> |Setting|U-Net|UpperNet|MSSNet|
> |-|-|-|-|
> |Original|50.12|49.77|52.06|
> |+OSM|54.86|53.77|55.54|
>
> \[\*1\] Herfort, B., Lautenbach, S., Porto de Albuquerque, J., Anderson, J., & Zipf, A. (2023). A spatio-temporal analysis investigating completeness and inequalities of global urban building data in OpenStreetMap. *Nature Communications*, 14, 3985
>
> \[\*2\] Du S, Du S, Liu B, et al. Mapping large-scale and fine-grained urban functional zones from VHR images using a multi-scale semantic segmentation network and object based approach. *Remote Sensing of Environment*, 261 (2021): 112480.

---

> > ### Comment · Reviewer_6XcF · 2025-08-05
> >
> > Thank you.
> >
> > The author addressed my questions. But I think the technical contribution of this work is still rather limited.

---

> > > ### Author Response · Authors · 2025-08-05
> > > **Thank you for your comment**
> > >
> > > Thank you very much for taking the time to review our paper and for your thoughtful comments. We truly respect your assessment and appreciate the opportunity to improve our work based on your feedback.
> > >
> > > While we understand your concern regarding the level of technical contribution, we would like to kindly highlight that GeoLink introduces a novel paradigm for integrating OSM data into RS FM pretraining. Specifically:
> > > 1. It enhances RS FM pretraining by leveraging cross-modal spatial correlations and multi-granularity learning signals derived from OSM, leading to consistent improvements over existing RS FMs on classification, segmentation, and other tasks;
> > > 2. It introduces a multimodal masked input strategy that improves pretraining efficiency;
> > > 3. It enables stronger performance and richer diversity on downstream tasks, significantly improving the effectiveness of a wide range of real-world applications, such as urban functional zone mapping.
> > >
> > > We hope these technical contributions can serve as a step forward in bridging RS and open geographic data, and be useful to both research and practical applications.
> > >
> > > Thank you again for your comments and the opportunity to clarify the motivation and technical contributions of our work.

---

### Official Review · Reviewer_FjL2 · 2025-06-28

**Clarity:** 3
**Significance:** 2
**Originality:** 2
**Rating:** 5
**Confidence:** 4

**Summary:**

This paper proposes GeoLink, a remote sensing (RS) foundation model that combines RGB images with OpenStreetMap (OSM) data. The model consists of two separate encoders for each data modality (RS images and OSM), a fusion strategy for combining OSM and image embeddings, and a decoder for the image modality. The pre-training strategy follows a Masked Autoencoder (MAE)-like approach. Results are reported on various datasets within the Earth Observation domain, covering downstream tasks such as classification, segmentation, and urban function zone mapping. The proposed approach demonstrates superior performance compared to related works, highlighting the value of incorporating OSM data during pre-training.

**Questions:**

- Why does the model not consider multispectral data, such as Sentinel-2? What is the motivation for limiting the RS encoder to RGB images? Given the wide availability and importance of multispectral data in Earth Observation (EO), it would be helpful to understand the reasoning behind this design choice. Are there specific challenges or limitations that prevent the integration of multispectral inputs into the current framework?


- How difficult is the overall process of obtaining OSM data? While OSM provides rich geospatial information, it is not as commonly used in EO foundation models as other modalities (e.g., text, multi-spectral, SAR). Could the authors elaborate on the practical challenges of collecting, aligning, or processing OSM data at scale? Are there domain or geographic limitations that affect its applicability?


- Why has the model been pre-trained for only 60 epochs? Is this decision based on empirical results, resource limitations, or performance? It would be useful to understand whether longer pre-training might lead to further gains, or if 60 epochs was sufficient based on validation trends.


- Regarding the ablation studies, what would be the expected impact of increasing the masking ratio for the RGB encoder (e.g., 80–90%)? Did the authors experiment with or consider more aggressive masking? If so, what were the observed effects?

**Ethical Concerns:**

["NO or VERY MINOR ethics concerns only"]

**Final Justification:**

The authors have properly addressed all of my concerns with both experimental evidence and thoughtful explanations, as demonstrated in their detailed rebuttal. I still believe that future work, such as the incorporation of multispectral data, could make the publication even stronger. However, despite this, the paper has merit, and I believe it would be a valuable contribution to NeurIPS. Therefore, I am increasing my previous rating to accept.

**Limitations:**

Yes

**Quality:**

3

**Strengths And Weaknesses:**

Strengths:

The paper is overall well written, which makes it enjoyable to read. However, in some parts (mostly the abstract, introduction, and conclusion), I noticed sentences that might contain patterns commonly found in AI-generated text, such as the overuse of hyphens. Note that I’m not claiming this is AI-generate, is just that the style sometimes feels unusual.

The novelty is somewhat limited in terms of the model’s components, as most are drawn from existing architectures (e.g., RS encoder: ViT, Mask Decoder: MAE, OSM encoder: GATConv). However, I found the integration of these elements to be valuable and clever.

The paper includes extensive experiments to demonstrate the value of the proposed approach. The model is evaluated on classification, segmentation, and urban function zone mapping downstream tasks. It shows superior performance on most datasets for each task when compared to related works. Additionally, ablation studies are provided to highlight the contribution of each model component.

This doesn't influence my decision, but I want to highlight it: the appendix is very informative. It contains detailed and useful descriptions of many aspects mentioned briefly in the main paper, including the model's components, downstream tasks, fine-tuning process, and implementation details.

Weaknesses:

I found the comparison a bit limited. Some of the works included, such as Scale-MAE or Cross-Scale MAE, do not explicitly incorporate multimodality. Since this is a multimodal approach, I believe it would be more appropriate to compare with recent models that utilise other types of modalities. For example: MMEarth and MultiMAE meets EO. Additionally, you might consider works that incorporate textual modalities. A broader perspective can be found in this recent survey: https://arxiv.org/pdf/2505.14361.

I don’t fully understand the motivation for including only RGB images in this domain. While RGB is the most abundant data type in EO, many datasets include multispectral data (e.g., Sentinel-2). Evaluating such data would make the submission stronger and more applicable to real-world EO scenarios, where S2 data is widely used. I understand that some recent EO models (e.g., CLIP-based) are limited to RGB due to the constraints of pre-trained models. However, in this case, the encoder is not fixed to such limitations. I’m not against using only RGB, nor am I criticising this point directly, but the paper lacks a convincing motivation for this design choice.

Some minor formatting errors and suggestions:
- I’d suggest placing Fig.2 on page 3. I think it will help the reader to have the figure in the same page where the description of the approach starts.
- Fig 2. Part (a). I’d suggest changing the abbreviation of projection to ‘proj’ instead of ‘prj’ on the blue box under ‘contrastive loss’. I agree that both abbreviations are correct, however since this is a computer vision paper, I believe ‘proj’ is more common.
- Line 139: Missing space between text and equation.
- Line 166: I’d suggest adding a reference to set2set layers (https://arxiv.org/pdf/1511.06391). Not sure if the term is universally known.
- Line 226: Six benchmarks are mentioned but only five examples given.
- I’d suggest including the numbers on each bar of Figure 3 (something like Figure 4).  Probably, it would be more valuable to present these numbers in a table format (?).
- Inconsistent capitalisation in section headers. For instance, line 112 uses "Framework Overview" while line 97 has "Synergy of RS and geospatial vector data." I recommend adopting a consistent style throughout the manuscript.

---

> ### Author Rebuttal · Authors · 2025-07-30
>
> We are very grateful to your insightful comments. Thank you for recognizing the strengths of our work: clear writing, novel framework, and good evaluation. Your suggestions regarding the multispectral EO data and baseline experiments are also highly valuable and will help us improve the quality of the paper. Please find our point-by-point responses below.
> ### **Q1. The use of multispectral EO data**
>
> Thanks for pointing this out. We agree that the inclusion of multispectral EO images could enhance the model. However, we would like to emphasize that the focus of this paper is to explore how to leverage the rich geographic information of OSM data in RS FM, not to include as much data as possible. Due to the challenges in OSM vector encoding and large domain gap, the use of OSM vector data in RS FMs is under-explored. **This paper aims to bridge this gap and demonstrate the value of OSM in improving the performance and task diversity of RS FM, not the designing of multispectral RS encoder.**
>
> Moreover, **GeoLink can be readily extended to support multispectral RS images by modifying the image encoder**. In its current design, GeoLink's fusion mechanism imposes no specific constraints on the image encoder. Therefore, in principle, we can replace the existing ViT-based encoder with any encoder capable of handling multispectral data, without modifying other key components of the architecture. Previous multispectral pretraining approaches—using either separate encoders (e.g., SkySense) or joint tokenization (e.g., DOFA)—offer promising ways for adapting GeoLink to multispectral data.
>
> **As mentioned in the conclusion section (Lines 345-347), making GeoLink to support multispectral inputs is an important direction**, and we are excited to explore this further in future work.
>
> ### **Q2. Baseline model settings**
> Thank you for your insightful suggestion. **To the best of our knowledge, there is currently no RS FM that uses the exact same data types as GeoLink**, making it difficult to directly compare with fully modality-aligned baselines. The starting point of our experimental design is to explore how, and to what extent, integrating OSM data can enhance RS FM.
>
> To fairly and comprehensively evaluate GeoLink, we consider baselines designed from different perspectives:
> * Unimodal RS FMs that use the same RS spectral modality as GeoLink, including GASSL, ScaleMAE, and Cross-scale MAE;
> * Multimodal models that incorporate additional modalities, such as CROMA and DOFA. **Following your suggestion, we find that MMEarth and MultiMAE Meets EO are indeed suitable baselines for GeoLink, as they incorporate other modalities in their designs, like land cover maps, which are similar to OSM data from the perspective of information content.** We have included MMEarth as a baseline for GeoLink in our updated comparisons, and as for MultiMAE Meets EO, we do not find its public checkpoints currently.
>
> We also replace DOFA-B by DOFA-L for comparison. The table below shows the average performance across multiple benchmarks for each task, where GeoLink consistently outperforms MMEarth and DOFA-L.
>
> |Model|Cls (kNN)|Cls (Linear)|Cls (FT)|Seg (Freezing)|Seg (FT)|CD (Freezing)|CD (FT)|
> |-|-|-|-|-|-|-|- |
> |MMEarth|83.32|90.18|92.73|46.99|52.91|59.37|61.29|
> |DOFA-L| 84.67|91.24|93.16|48.07|54.94|60.12|61.95|
> |GeoLink|87.06|92.60|94.53|51.88|56.11|61.58|63.00|
>
> Moreover, **we believe it is not appropriate and fair to compare GeoLink with vision-language models (VLM) that integrate textual information, because they are fundamentally different:**
> * As noted in the review article \[\*1\] you list, most RS VLMs are not trained from scratch but are finetuned from VLMs from computer vision field like CLIP, which are pretrained on massive natural images (e.g., 400M image-text pairs for CLIP pretraining). In contrast, RS FMs are generally trained from scratch and VLMs are not used as baselines in previous RS FM works \[3, 10, 32, 33, 34\] as well.
> * VLMs are typically evaluated in zero-shot settings and focus on aligning images with textual semantics, which is fundamentally different from GeoLink's motivation—leveraging freely available geospatial data to improve RS FM.
>
>
> \[\*1\] Weng X, Pang C, Xia G S. Vision-Language Modeling Meets Remote Sensing: Models, datasets, and perspectives. IEEE Geoscience and Remote Sensing Magazine, 2025.
>
> ### **Q3. The process of obtaining OSM data**
> Thank you for the insightful question. We would like to clarify that **obtaining OSM data is relatively straightforward and scalable**. The OSM data is openly accessible through public APIs (e.g., Overpass) and bulk downloads (e.g., Geofabrik). **Our data processing pipeline is fully automated** to extract relevant OSM vector features for any given region. Specifically:
> 1. We use the geolocation coordinates and timestamps of RS images to query the corresponding OSM data via the Overpass API;
> 2. The OSM objects are clipped using spatial overlay to align them accurately with the RS images, forming image-OSM sample pairs;
> 3. We use rule-based cleaning to fix common issues—like invalid polygons (e.g., self-intersections), duplicate or conflicting objects, and tags with formatting or spelling errors. This helps correct errors and ensures OSM inputs meet GeoLink's structural needs.
> 4. After encoding the tags and computing spatial relationships, we construct OSM graphs as input to the model.
>
> In our experience, while many studies highlight the importance of OSM in GeoAI \[19\] , its practical usage remains limited—**mainly due to challenges in effectively encoding its semantic and geometric information and integrating it with EO data.** GeoLink is designed to address these bottlenecks.
>
> Additionally, as you suspect, despite its many advantages, OSM data does have limitations in practice. One key concern is its **uneven global coverage (e.g., better in developed urban centers than in rural areas)**, which may introduce geographic bias (which is an objectively common issue in FMs \[19\]) when used for pretraining FMs. To address this, one can explore the following directions:
>
> **Data-side**: Mitigating the systematical bias by increasing sampling density in OSM-sparse regions;
>
> **Model-side**: Developing globally location-aware FMs that treat geographic space as a continuous input to guide the modeling of priori knowledge of geographic distribution, as explored in \[42\].
> ### **Q4. Pretraining epochs**
> Thank you for your thoughtful observation. The decision to pretrain GeoLink for 60 epochs is driven by a combination of empirical findings and resource constraints. **We find that training for more epochs (e.g., 100) lead to only marginal performance gains, indicating its convergence around epoch 60.** Given the limitation of computational resources, we choose 60 as a practical default.
> ### **Q5. Masking ratio for image encoder**
> Thank you for the helpful suggestion. We experiment with masking ratios of 80% and 90%. As shown in the table below, we observe a slight performance drop at 80% and a significant collapse at 90%. This result is consistent with observations from the original MAE paper. Therefore, we adopt a 75% masking ratio for image patches in our study.
>
> |Mask Ratio|Cost|kNN|Linear|UFZ-U|UFZ-M|
> |-|-|-|-|-|-|
> |75%(Default)|1.0×|87.06| 92.60|55.40|60.00|
> |80%|0.95×|85.21|90.17|52.01|56.04|
> |90%|0.9×|78.02|83.47|43.02|45.81|
>
> ### **Q6. Minor formatting errors**
> We sincerely appreciate your kind reminder. Due to NeurIPS policy, we cannot update the manuscript during the rebuttal phase. And **we will carefully revise the final version to address the formatting and writing issues you kindly pointed out**. Thank you again!

---

> > ### Comment · Reviewer_FjL2 · 2025-08-02
> >
> > Many thanks to the authors for their insightful replies to each of my comments. I really appreciated that you ran extra experiments to address some of my concerns. Would it be possible to add those discussions to the paper? In particular, please focus on the motivation for using only RGB images and the experiments that extend the comparison against prior related work.
> >
> > Regarding the comparison with VLMs, this is okay. I understand that it would not be fair to compare against those works. However, it would be an interesting exercise to evaluate the contributions of different modalities. Anyways, I fully agree that this does not need to be included in the paper.
> >
> > Experiments with masking ratios could be detailed in the supplemental material. In the main paper, it would be enough to simply mention that different masking ratios were used and that a particular ratio led to better results.
> >
> > Many thanks again.

---

> > > ### Author Response · Authors · 2025-08-03
> > > **Thank you for your comment**
> > >
> > > Many thanks for your kind words and constructive feedback. We believe it was our responsibility to run the additional experiments to properly address your concerns, and we truly appreciate the opportunity to clarify these points.
> > >
> > > We will certainly incorporate the discussions from the comments and responses into the final version of the paper, including:
> > > 1. The motivation for using only RGB images: we will revise the Introduction and Experiments Sections of the paper to further clarify and emphasize the motivation for using only RGB image data.
> > > 2. Adding MMEarth and DOFA-L as comparison baselines: in the Experiments Section, we will include MMEarth as a baseline method for comparison, and replace DOFA-B with DOFA-L to ensure a fairer evaluation.
> > > 3. Image masking ratio: this ablation study will be added to the supplemental material, with a summary included in the main paper.
> > > 4. The processing of OSM data: combining your comments with Reviewer uK7F, we will incorporate the processing of OSM data into the paper and supplemental material.
> > > 5. Minor formatting errors: Thank you again for your detailed and thoughtful review, which pointed out several minor errors in our paper. We will correct them in the final version.
> > >
> > > According to the NeurIPS 2025 guidelines, we are unable to update the manuscript at this stage, and all the proposed revisions will be incorporated into the final version.
> > >
> > > Moreover, we appreciate your understanding about the comparison with VLMs. We fully agree that evaluating the contributions of different modalities in RS and geospatial tasks is an interesting and valuable exercise. In future work, we will explore how to design suitable experiments to investigate this, as you suggested.
> > >
> > > Once again, thank you for your valuable feedback and the time you dedicated to reviewing our work. Your insightful suggestions have greatly helped improve the clarity and completeness of the paper!

---

### Official Review · Reviewer_uK7F · 2025-07-02

**Clarity:** 3
**Significance:** 3
**Originality:** 2
**Rating:** 5
**Confidence:** 5

**Summary:**

The paper presents an approach to linking remote sensing imagery with OSM data, taking into account the different data structures. The result is OSM enhancing remote sensing data. In addition, the method addresses computational issues by enabling sparse input.  The method is applied to segmentation tasks to test for improvement. The idea is interesting and nicely presented.

**Questions:**

How would this used in practice? i.e. how does one obtain a RS and OSM pair that are temporally aligned and good data?

How is the accuracy of the OSM data dealt with?

**Ethical Concerns:**

["NO or VERY MINOR ethics concerns only"]

**Final Justification:**

The authors have responded well during the discussion. I am happy with the responses and the proposed edits to the final version of the paper. My scores remain the same.

**Limitations:**

These have been addressed.

**Quality:**

3

**Strengths And Weaknesses:**

ln 27: fix the language
ln 32-33: This statement isn't really true - revise please.
Captions of figures should be in the figure environment not in the graphic itself.

There are important results in the supplementary as well that should perhaps be included in the main part of the paper?

---

> ### Author Rebuttal · Authors · 2025-07-30
>
> We are very grateful for your insightful comments. We also sincerely appreciate your comprehensive and accurate summary of our work, as well as your recognition of the novelty and writing clarity. Your suggestions are truly valuable and will greatly help us improve the quality of this paper. Please find our point-by-point responses below.
>
> ### **Q1. Minor issues in language, statement, and graphics**
>
> Thank you for your kind reminder. We will revise the statement in Lines 32-33 to: "However, the integration of ground-level geospatial data remains relatively underexplored in many existing RS FMs...". And **we will carefully incorporate all your suggestions regarding writing and figure issues in the final version** of the paper since NeurIPS does not allow manuscript revisions during the rebuttal phase. We sincerely appreciate your thorougth review!
>
> ### **Q2. Important results in supplementary**
>
> Thank you for pointing this out. Due to the 9-page limit imposed by NeurIPS, we include several experimental results in the supplementary material—such as evaluations on the urban village task, performance under limited annotations, and impact of loss weight. As noted in the NeurIPS guidelines, the final version may include one additional content page. We plan to move some of these important results, such as the urban village evaluation and key ablations, into the main text to better highlight the strengths and characteristics of our approach.
>
> ### **Q3. The practical use of GeoLink**
>
> Thank you for your valuable and practical suggestion. **One can automatically obtain spatiotemporally aligned RS-OSM data pairs and then rely on robust post-processing methods like similarity check to ensure data quality.** We will explain this from two aspects.
>
> **Obtaining spatiotemporally aligned data pairs**:
> Each RS image used in this study contains metadata on geographic location and timestamp. Based on this information, we retrieve corresponding OSM data via the Overpass API, which **allows querying OSM data within a specified time range**. This processing pipeline is fully automated. Users of GeoLink can replicate this approach using open-access sources (e.g., Google Earth Engine) to obtain RS images, and then retrieve corresponding OSM data using the image's location and timestamp via Overpass or similar APIs.
>
> **Ensuring data quality**:
> We are aware that OSM data may contain noise due to outdated or inaccuracy annotations, which can result in cases where the OSM data is not sufficiently relevant to the corresponding RS image. To mitigate this, we recommend following the method in SkyScript \[29\], where OSM tags are converted into natural language descriptions and vision-language models, like CLIP, are used to **calculate the similarity between RS images and OSM-based textual descriptions.** Low-similarity pairs can then be filtered out to obtain higher-confidence, spatiotemporally aligned samples.
>
> As stated in the Checklist, we will open-source the data processing code after the blind review to support reproducibility and ease of use.
>
> ### **Q4. How to ensure the accuracy of OSM data in GeoLink**
>
> Thank you very much for raising this important point. As you mentioned, OSM data collected via crowdsourcing can have quality issues such as outdated or inaccurate annotations. **In this work, we mitigate the impact of OSM noise through data cleaning, sampling-based quality checks, and model robustness:**
>
> **Data cleaning**:
> After retrieving OSM data via Overpass, we apply rule-based cleaning to remove common issues—such as fixing or removing invalid polygons (e.g., self-intersections), eliminating duplicate/conflicting objects, and filtering tags with formatting or spelling errors. These steps not only correct obvious errors but also ensure that OSM inputs meet structural requirements for the model. Additionally, the pretraining data used in GeoLink is derived from SkyScript \[29\], which verifies the match quality between RS images and temporally aligned OSM data and ranks the match confidence. We only use the top 30% ranked RS-OSM pairs for pretraining, further ensuring alignment quality.
>
> **Sampling-based validation**:
> We manually sample both pretraining and downstream datasets to inspect OSM quality and its alignment with RS images. From a random sample of 1,000 RS–OSM pairs, less than 0.02% of OSM objects show quality issues.
>
> **Model robustness**:
> **GeoLink is designed with noisy OSM data in mind.**
> First, its use of natural language encoding reduces sensitivity to the tag noise. For example, if a *reservoir* is mislabeled as a *lake* in OSM, GeoLink may still obtain some useful supervision during pretraining due to the semantic similarity between "reservoir" and "lake".
> Second, the OSM graph encoder uses topological relationships to describe spatial structure as edges, which reduces the impact of geometric inaccuracies and further improves robustness.
>
> In summary, GeoLink addresses the potential accuracy limitations of OSM data from multiple angles. As demonstrated by our experimental results, despite the real-world noise in OSM data, its integration still enhances the performance of RS FM during both pretraining and downstream tasks.

---

> > ### Comment · Reviewer_uK7F · 2025-08-01
> >
> > Thank you for the thorough explanations. Can this be incorporated into the paper?

---

> > > ### Author Response · Authors · 2025-08-02
> > > **Thank you for your comment**
> > >
> > > Of course. We sincerely appreciate all of your insightful suggestions and will incorporate them into the final version of the paper. We believe these improvements will enhance both the clarity and practical value of GeoLink, making it easier for readers to understand and apply.
> > >
> > > Thank you again for your constructive feedback and the time and effort you dedicated to reviewing our work!

---

### Official Review · Reviewer_gYwk · 2025-07-04

**Clarity:** 3
**Significance:** 3
**Originality:** 3
**Rating:** 5
**Confidence:** 3

**Summary:**

This paper proposes GeoLink, a framework to leverage OSM data to enhance Remote Sensing Foundation model. To combine these two modalities, GeoLink uses self-supervised mask reconstruction learning for efficient pretraining. GeoLink is then tested across multiple unimodal tasks and datasets, and multimodal tasks, showing improvements compared to baselines.

**Questions:**

* There have been quite a few more recent geospatial foundation model that the work should compare to: https://arxiv.org/pdf/2410.22629 If such models are not applicable, please let us know so and why). I may increase the score based on the results. I will decrease the score if no justification or results were given.
* For Table 1, the paper only compares to DOFA VIT-B. In the DOFA paper, it shows their DOFA VIT-L has slightly better performance. Please include VIT-L as a baseline too or provide a justification on why it shouldn't be. I may increase the score based on the results. I will decrease the score if no justification or results were given.
* For the multimodal tasks experiments, please include task-specific baselines too. This will help the community understand when to use foundation models vs. task-specific ones. I may increase the score based on the results. I will decrease the score if no justification or results were given.

**Ethical Concerns:**

["NO or VERY MINOR ethics concerns only"]

**Final Justification:**

Based on the rebuttal, I have raised the score. In general, the work is a very nice contribution to the community to empower better Geospatial reasoning, with strong and in-depth experiments and insights.
The rebuttal resolved my concern on adding more geospatial foundation models, adding VIT-L as benchmark, adding task-specific baselines.

**Limitations:**

yes

**Quality:**

4

**Strengths And Weaknesses:**

Positives
* Well-written paper that does well showcasing the intuition and justification for design decisions
* The idea of doing image mask-reconstruction with OSM and remote sensing image is intuitive and sound and allows leveraging of OSM data in RS.
* Good improvement on four tasks, the diversity of task is also good (spanning unimodal and multimodal vision tasks)
* I especially appreciate the OSM data completeness table which adds practical insight since OSM is not always complete.

Areas of improvement
* There have been quite a few more recent geospatial foundation model that the work should compare to: https://arxiv.org/pdf/2410.22629 If such models are not applicable, please let us know so and why).
* For Table 1, the paper only compares to DOFA VIT-B. In the DOFA paper, it shows their DOFA VIT-L has slightly better performance. Please include VIT-L as a baseline too or provide a justification on why it shouldn't be.
* For the multimodal tasks experiments, please include task-specific baselines too. This will help the community understand when to use foundation models vs. task-specific ones.

---

> ### Author Rebuttal · Authors · 2025-07-30
>
> We are very appreciated to your insightful comments. Thank you for recognizing the strengths of our work: clear writing, sound methodology, and thorough experiments. Your suggestions regarding the baseline experiments are also highly valuable and will help us improve the quality of the paper. Please find our point-by-point responses below.
>
> ### **Q1. Comparison with CrossEarth**
>
> Thank you for your kind reminder. **Since CrossEarth is a vision FM specifically designed for RS semantic segmentation under cross-domain generalization, it can also be applied to some tasks in GeoLink for comparison.** To ensure a fair and reasonable evaluation, our experimental design considers the following two aspects:
>
> **The selection of CrossEarth version:**
> CrossEarth provides multiple domain generalization scenarios (e.g., urban-to-rural, different climate zones) and releases models trained on various source datasets. Based on our experimental setup and the available open-source models, we select the versions trained on ISPRS Potsdam (RGB) and CASID (Temperate Monsoon), as both are closely related to land cover tasks, similar to our two benchmarks on semantic segmentation task: FiveBillionPixels and AI4Smallfarms. This setting helps reduce the impact of domain gaps and better reflect the true performance of CrossEarth.
>
> **Adaptation setting:**
> In CrossEarth's original zero-shot setting, the source and target domains share the same label taxonomy. However, this is not the case for our benchmarks. Therefore, we modify the classification head of CrossEarth to match the number of classes in our benchmarks, while keeping all other layers and pretrained weights fixed. We finetune the classification head on the target datasets and compare the results to GeoLink with freezing encoder. This setting helps CrossEarth adapt to our benchmarks and preserve its knowledge learned from the source domain.
>
> Experimental results in the table show that CrossEarth exhibits strong generalization ability and performs reasonably well, although slightly below GeoLink. Notably, the model pretrained on CASID outperforms the one pretrained on Potsdam, possibly due to a closer distributional similarity to our benchmark datasets. This comparison highlights both CrossEarth's strength as a domain-generalized RS segmentation FM and GeoLink's effectiveness in leveraging OSM data to enhance the generalization of RS FM. In addition, FMs like CrossEarth, which focus on a specific type of downstream task, offer valuable insights for enabling more efficient adaptation of GeoLink in downstream applications. Combining techniques such as SSL pretraining and domain generalization can further help reduce the annotation requirements of FMs. We will include and discuss this experiment in the final version.
>
> |Model|Source domain|FiveBillionPixels|AI4Smallfarms|
> |-|-|-|-|
> |CrossEarth|ISPRS Potsdam|54.33|38.24|
> |CrossEarth|CASID|56.18| 41.07|
> |GeoLink|/| 60.49| 43.26|
>
> ### **Q2. Comparison with DOFA-ViT-L**
>
> Thank you for your thoughtful suggestion. We have **replaced DOFA-B with DOFA-L** in our experiments as the basline and re-evaluated the results.In addition, we add another recent RS FM MMEarth \[*1\] for comparison. As shown in the table below, we present the average performance of DOFA-B, DOFA-L, MMEarth, and GeoLink across multiple benchmarks for each kind of task. DOFA-L shows clear improvements over DOFA-B, and GeoLink still outperforms DOFA-L in all tasks. We will update the baseline models in the final version of the manuscript.
>
> |Model|Cls(kNN)|Cls (Linear)|Cls (FT)|Seg (Freezing)|Seg (FT)|CD (Freezing)|CD (FT)|
> |-|-|-|-|-|-|-|-|
> |DOFA-B|81.66|90.66|92.17|47.67|54.04|59.55|61.54|
> |DOFA-L|84.67|91.24|93.16|48.07|54.94|60.12|61.95|
> |MMEarth|83.32|90.18|92.73|46.99|52.91|59.37|61.29|
> |GeoLink|87.06|92.60|94.53|51.88|56.11|61.58|63.00|
>
>
>
> ### **Q3. Comparison with task-specific baselines**
>
> Thank you for your valuable and practical suggestion. Based on your suggestion and the real-world application scenarios in the RS/geospatial community, we evaluate task-specific models for multimodal urban functional zone (UFZ) segmentation from two perspectives: performance under limited annotations and geographic generalization. **GeoLink outperforms task-specific models in both limited-sample and cross-region generalization settings, highlighting its strength in multimodal application scenarios.**
>
> **Task-specific baseline setting:**
> To the best of our knowledge, there are very few open-source task-specific models in the RS/geospatial domain that use the same types of data as GeoLink. Some proposed methods in the published literature are also difficult to reproduce reliably only based on the descriptions provided. The survey \[\*2\] offers a detailed categorization and summary of existing methods for UFZ segmentation. Following its guidance, we designed several representative task-specific models for comparison.
> (1) **U-Net+OSM**: using ImageNet-pretrained ResNet50 as the backbone to represent convolutional models. Since the fusion module in GeoLink is plug-and-play (Lines 646-648), we use the same OSM encoder and fusion encoder from GeoLink to ensure a fair comparison (also applied to the models below).
> (2) **UpperNet+OSM**: using ImageNet-pretrained ViT-L as the backbone, representing Transformer-based architectures.
> (3) **MSSNet+OSM**: MSSNet \[\*3\] is a task-specific model designed with multi-scale mechanisms tailored for UFZ segmentation.
>
> **Performance under limited annotations:**
> We test model performance using 10\%, 50\%, and 100\% of UFZ training data. As shown in the following table, compared with the task-specific models, GeoLink performs significantly better with fewer samples. GeoLink using only 10% of the data outperforms these models even when trained with 50% of the data. The performance gap between GeoLink and the task-specific models is largest when using only 10\% of data, and narrows as more annotations becomes available. Among the task-specific models, MSSNet performs better than U-Net and UpperNet, suggesting that task-specific architectural design can be beneficial in some scenarios.
>
> |Data Ratio|U-Net+OSM|UpperNet+OSM|MSSNet+OSM|GeoLink (multimodal)|
> |-|-|-|-|-|
> |10\%|41.28|42.97|43.09|53.12|
> |50\%|50.01|49.09|51.96|56.95|
> |100\%|54.86| 53.77| 55.54|60.00|
>
> **Geographic transferability:**
> Our UFZ benchmark includes cities with highly distinct geographic conditions: Singapore, Chicago (US), and Shenzhen (China). The original paper uses a mixed training/testing setting. To evaluate transferability, we train models on one city and test on another. Specifically, we train GeoLink on Singapore and test on Shenzhen and Chicago. As shown in the table, although all models experience performance drops due to domain shift, GeoLink consistently outperform the task-specific models in the target cities, highlighting its superior generalization across regions. This demonstrates the effectiveness of pretrained FMs in capturing transferable geographic features.
>
> |Source Region|Target Region|U-Net+OSM|UpperNet+OSM|MSSNet+OSM|GeoLink (multimodal)|
> |-|-|-|-|-|-|
> |Singapore|Singapore|54.39|53.82|56.87|61.82|
> |Singapore|Shenzhen|32.51|32.21|33.23|37.25|
> |Singapore|Chicago|39.62|40.06|41.07|49.12|
>
> **Discussion: These results suggest that the choice between RS FMs and task-specific models should depend on training data availability and region scale.** FMs, pretrained on large-scale unlabeled RS data, offer stronger generalization and are well-suited for low-resource or large-scale, cross-region mapping tasks. As for task-specific models, they can often be more flexibly customized in terms of model architecture.When designed carefully, task-specific models can perform competitively with sufficient training data. Furthermore, combining task-specific model design with SSL pretraining on unlabeled data may provide a strong initialization and lead to both task alignment and generalization.
> In the future, we envision RS FMs becoming more flexible and capable of integrating more modalities—including but not limited to RS images, geospatial vectors, and street-level data—to support an even broader range of geographic tasks.
>
> \[\*1\] Nedungadi V, Kariryaa A, Oehmcke S, et al. Mmearth: Exploring multi-modal pretext tasks for geospatial representation learning. *European Conference on Computer Vision*. Cham: Springer Nature Switzerland, 2024: 164-182.
>
> \[\*2\] Li Z, Chen B, Wu S, et al. Deep learning for urban land use category classification: A review and experimental assessment. *Remote Sensing of Environment*, 311 (2025).
>
> \[\*3\] Du S, Du S, Liu B, et al. Mapping large-scale and fine-grained urban functional zones from VHR images using a multi-scale semantic segmentation network and object based approach. *Remote Sensing of Environment*, 261 (2021): 112480.

---

> > ### Comment · Reviewer_gYwk · 2025-08-04
> >
> > Dear authors, thank you for your rebuttal which included many experiments to address the concerns I have brought up, especially on the additional FM baselines and also task-specific baselines.
> > Please include the above experiments and changes to the final paper. Please also add your discussion point onto when to use FM vs. task-specific baseline. This is very interesting!

---

> > > ### Author Response · Authors · 2025-08-05
> > > **Thank you for your comment**
> > >
> > > Thank you very much for your thoughtful and encouraging feedback.  In particular, your suggestion to compare FMs with task-specific models was highly insightful — it not only guided us to conduct meaningful additional experiments, but also significantly improved the practical value of the work.
> > >
> > > We will definitely incorporate the results of these new experiments and discussions into the final version of the paper, including:
> > > 1. Adding CrossEarth, MMEarth, and DOFA-L as comparison baselines.
> > > 2. Adding the task-specific models as the baselines for the multimodal tasks.
> > > 3. Including the discussion on when it is more appropriate to use a FM versus a task-specific model.
> > >
> > > According to the NeurIPS 2025 guidelines, we are unable to update the manuscript at this stage, and all the proposed revisions will be incorporated into the final version.
> > >
> > > Once again,we sincerely appreciate the time and effort you devoted to reviewing our paper. Your insightful suggestions have greatly contributed to improving both the quality and the real-world relevance of our work！

---

### Note · Authors · 2025-08-11

We would like to express our sincere gratitude to the reviewers for your valuable feedback, and to the AC and SAC for your careful coordination and dedication throughout the review process.

We are glad that the reviewers think our paper is well presented (Reviewers gYwk, uK7F, FjL2, BhhX); our approach is sound, interesting, and promising (Reviewers gYwk, uK7F, FjL2, 6XcF, BhhX); our paper proposes a relatively novel framework (Reviewers 6XcF, FjL2); our proposed method has been well tested with extensive experiments and achieves superior results (Reviewers gYwk, FjL2, 6XcF).

The main contributions of the paper can be summarized as follows:
1. **Technical perspective:** We propose GeoLink, a multimodal FM that enhances RS self-supervised pretraining through OSM-derived multi-granularity learning signals and enables efficient pretraining via masked input.
2. **Application perspective:** GeoLink demonstrates strong performance in real-world tasks such as urban functional zone segmentation, showcasing its capability to integrate multi-source information for tackling complex applications.
3. **Geoscience perspective:** Unlike previous approaches based on data conversion, data derivation, or knowledge graph, GeoLink explores a new and feasible way to mine knowledge from massive open-source crowdsourced geographic data to serve geoscientific research.

The main concerns raised by the reviewers are:
1. The quality and processing of OSM data.
2. The practical use and choices of GeoLink.
3. The use of multispectral RS images and comparison with CROMA.
4. The designing of OSM encoder.
5. The computational scalability and inference efficiency.

We have addressed the reviewers' questions one by one and will revise the paper to incorporate the reviewers' feedback, mainly including:

1. Adding the acquisition process of OSM data to guide the practical use.
2. Adding baselines, including CrossEarth, MMEarth, and DOFA-L.
3. Clarifying our motivation for using RGB RS images in the Introduction and Experiments sections.
4. Incorporating additional ablation studies, including image masking ratio, OSM encoder variants, and language supervision.
5. Discussing on when to use FMs vs. task-specific models, RS FM evaluation complexities, and the computational scalability and inference efficiency.
6. Correcting all minor formatting errors.

Many thanks again to all reviewers, the AC, and the SAC for your time and dedication!

---

### Decision · Program_Chairs · 2025-09-17

**Decision:**

Accept (poster)

**Comment:**

GeoLink incorporates the public data available through OpenStreetMap (OSM) into the self-supervised pre-training of remote sensing models. The technical contributions include the joint model of image and graph data, the data collection and filtering process for the graph data from OSM to make it reliable for self-supervision, and the design of the multi-scale losses across the imagery and OSM data. The experiments cover the use of OSM data upstream, during pre-training with multi-scale masking and contrasting tasks across image data from satellites and graph data from OSM, and downstream, during deployment by fusing imagery and the OSM data. The evaluations cover multiple remote sensing tasks (agriculture, urban land use and functional zones, disaster management, population, and carbon emissions) formulated as semantic segmentation, change detection, and regression. The proposed full model improves accuracy when transferring features and fine-tuning compared to multi-scale baselines (Scale-MAE, Cross-Scale MAE) and multi-modal baselines (CROMA, DOFA).

Five reviewers with expertise on vision, self-supervision, remote sensing vote for acceptance (gYwk: 5, uK7F: 5, FjL2: 5, BhhX: 4) and rejection (6XcF: 3). The review by uK7F is short and non-technical, although the reviewer did engage with the rebuttal, so this review has been downweighted in the decision. The borderline scores are the result of detailed evaluations and discussion and are maintained so these are fully considered in the decision. These final scores followed the rebuttal by the authors, which resulted in raised (gYwk, FjL2) and maintained scores. The area chair notes that the review process has further improved the submission with additional details on the collection and processing of the OSM data, additional and more recent baselines for multimodal remote sensing, discussion of limitations like the choice of RGB-only imagery and computation, and further ablations of the proposed multi-part framework. The final justifications of all reviewers except 6XcF confirm acceptance. The discussed modeling weakness of relying on over-simplified image input (just RGB, and not the multi-spectral images or other modalities common to remote sensing) can be reinterpreted as a strength by the counterpoint that with OSM data the model nevertheless performs satisfactorily. The remaining argument for rejection by 6XcF is insufficient technical novelty, but it is acknowledged that the rebuttal addressed the points raised by the review. The area chair agrees with the reviewer that the overall framework is novel, even if each component is not, and so on the balance this issue is resolved. The submission is informative, significant, and novel in its incorporation of the vector-format OSM data into deep learning for remote sensing.

After full inspection of the submission, the reviews, the rebuttal, and discussion the area chair sides with the majority of reviewers on acceptance.